# The Shaped Transformer: Attention Models in the Infinite Depth-and-Width Limit

**Lorenzo Noci**[*†]    **Chuning Li**[*‡]    **Mufan (Bill) Li**[*‡]    **Bobby He**[§]

**Thomas Hofmann**[†]    **Chris Maddison**[‡]    **Daniel M. Roy**[‡]

## Abstract

In deep learning theory, the covariance matrix of the representations serves as a proxy to examine the network's trainability. Motivated by the success of Transformers, we study the covariance matrix of a modified Softmax-based attention model with skip connections in the proportional limit of infinite-depth-and-width. We show that at initialization the limiting distribution can be described by a stochastic differential equation (SDE) indexed by the depth-to-width ratio. To achieve a well-defined stochastic limit, the Transformer's attention mechanism is modified by centering the Softmax output at identity, and scaling the Softmax logits by a width-dependent temperature parameter. We examine the stability of the network through the corresponding SDE, showing how the scale of both the drift and diffusion can be elegantly controlled with the aid of residual connections. The existence of a stable SDE implies that the covariance structure is well-behaved, even for very large depth and width, thus preventing the notorious issues of rank degeneracy in deep attention models. Finally, we show, through simulations, that the SDE provides a surprisingly good description of the corresponding finite-size model. We coin the name *shaped Transformer* for these architectural modifications.

## 1 Introduction

Pre-trained large language models have experienced a remarkable increase in popularity due to their eerily human-like ability to puzzle through complex reasoning tasks, solve coding challenges, and produce pages of logically sound text [1]. Arguably, the Transformer is the foundation of these successes [2]. Recent research has found evidence for scaling laws, linking the performance of these architectures to their parameter counts and the quantity of training data, fueling the desire to train deeper and wider models on ever larger datasets in order to unlock new levels of performance [3–7].

Bundled with the increased expressivity of deep architectures, however, is increased numerical instability, both in the forward pass and gradients, which hinders training. One of the clearest examples of instability is the so-called rank collapse phenomenon [8, 9] – the observation that, in Softmax-based attention models, the network's representation of different tokens tend to perfectly align at large depth. The resulting poorly conditioned covariance and correlation between tokens leads to exploding and/or vanishing gradients at initialization, disrupting gradient updates of the affected parameters. This situation violates a well-known guiding principle from the literature of deep signal propagation: a stable covariance is a necessary condition for stable training [10–15]. In fact,

---

[*]Equal contribution. Correspondence to:
`lorenzo.noci@inf.ethz.ch`, `chuning.li@mail.utoronto.ca`, `mufan.li@mail.utoronto.ca`
[†]ETH Zurich
[‡]University of Toronto and Vector Institute
[§]University of Oxford

37th Conference on Neural Information Processing Systems (NeurIPS 2023).

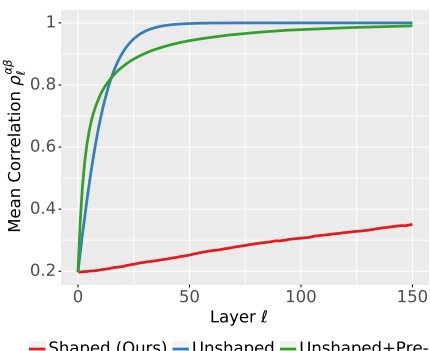 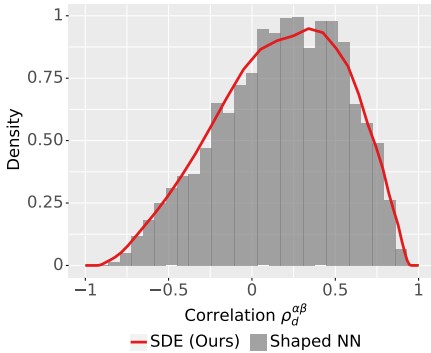

Figure 1: Our shaped Transformer prevents token representations from becoming perfectly aligned, i.e. rank collapse. Left: mean correlation $\rho_\ell^{\alpha\beta}$ of Transformers (Eq. 11) with and without shaped attention (Eq. 9) and Pre-LN [48]. Right: kernel density estimate and histogram of correlations from covariance SDE in Theorem 4.2 and shaped attention NN. Here we note correlation converging to 1 implies a poorly conditioned covariance matrix. Simulated with $n = 200, d = 150, \gamma = 1/\sqrt{8}, \tau_0 = 1, \rho_0^{\alpha\beta} = 0.2$, SDE step size $0.01$, and $2^{12}$ samples.

the instability of Transformers is evident when considering the critical role of hyperparameter tuning and the judicious use of normalization layers. In this work, we study Transformers in a novel infinite limit, rectify sources of instability with a novel modification, and derive the SDEs characterizing the covariance and output distribution.

Scaling limits have been used successfully to provide guidance on architecture [16–18] and tuning hyperparameters settings [19]. Our work represents a contribution in this direction. The ability to use such limits to diagnose instabilities depends on their tractability and faithfulness to real-world (finite) networks. In this regard, not all limits are created equal. In particular, the faithfulness of scaling limits depends critically on how other parameters are scaled with width. One of the simplest (and thus most popular) limits to work with – the "NTK" limit [20–24] – treats the depth of the network as fixed. As a result, at initialization, this limit does not accumulate sufficient random fluctuations over the depth of the network, leading to deterministic covariance matrices that do not agree with those of standard (finite) networks. Such networks have another defect: they are incapable of learning features in the limit [25]. Various other limits have been studied, towards identifying tractable yet faithful models of initialization and/or training. These include mean field limits [26–29] and the perturbative regime [30–36].

This work operates in a relatively new regime – the *proportional* infinite depth-and-width limit – where depth $d$ and width $n$ diverge as the ratio $d/n$ tends to a positive constant. This limit, first analyzed by Hanin and Nica [37], has been the recent subject of study in the context of neural network [38–41, 18]. A related line of work also studied the Lyapunov exponent for products of random matrices [42–45]. This regime retains the network's stochasticity and, at initialization, has been shown to closely resemble the behaviour of finite architectures, yet still yield a relatively simple limiting description, expressible in terms of stochastic differential equations [40, 18]. In this work, we fully characterize the initial output distributions of a network with skip connections and Softmax-based attention mechanisms, in the proportional infinite-depth-and-width limit.

Inspired by the idea of shaping activation functions [16–18, 46], our theoretical approach finds an adequately modified attention mechanism via its SDE limit. Our modification involves making the attention matrix closer to the identity, and appropriately choosing the temperature parameter $\tau$, which re-scales the logits of the Softmax. Similar to shaping activation functions, the temperature scaling we devise linearizes and reduces the saturation of the Softmax, a known source of training instability in Transformers [47]. In order to model the feedforward layer of a Transformer's block, we extend existing results [18] to derive an SDE for the proportional limit of shaped-ReLU feedforward multi-layer perceptrons (MLPs) with skip connections. Combined, we fully characterize the output distribution of a Transformer with shaped non-linearities (Corollary 4.3).

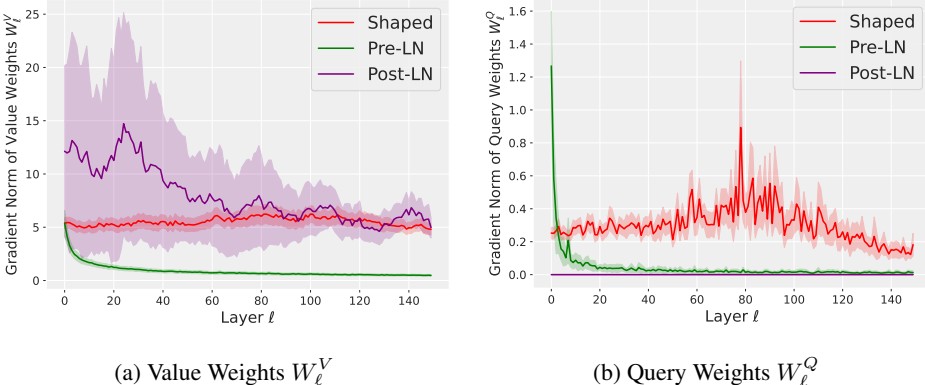

(a) Value Weights $W_\ell^V$              (b) Query Weights $W_\ell^Q$

Figure 2: Comparing gradients norms at initialization for different parameters as a function of depth, with and without shaped attention. The architecture is the same as in Figure 1 but with autoregressive causal masking, and the task is next-token prediction on code data. Left: Value weights $W_\ell^V$ for shaped attention, standard Pre-LN, and the original Post-LN block [2]. Right: the same gradient norm plot but for Query weights $W_l^Q$. We find that shaping the attention mechanism successfully prevents gradients from vanishing, while unshaped Transformers suffer from rapidly vanishing gradients. Interestingly, only the Post-LN query gradients vanish, but value gradients are stable across depths, which is consistent with the findings of Noci et al. [9]. On the other hand, shaped attention has stable gradients for both parameters inside and outside the Softmax nonlinearity.

Notably, our modification successfully prevents a poorly conditioned covariance matrix, whereas the vanilla Softmax-based attention model without LayerNorm [49] fails in this regard, and the corresponding Pre-LN architecture provides only marginal improvements (see Figure 1). Given that our modification is inspired by previous work on shaping activation functions, we coin the terms *shaped attention* for the proposed attention mechanism and *shaped Transformer* for the overall architecture that includes the MLP block and residual connections. Through simulations (e.g., Figure 1), we show that the limiting neural covariance SDE approximates the distribution of finite-size Transformers with shaped attention mechanism surprisingly well. We also provide preliminary training experiments for our proposed shaped attention architecture on standard language modeling tasks, demonstrating the feasibility of the new architecture in practice (see Section 5 and Appendix D).

In summary, our contributions are as follows:

1. We study the effect of skip connections in the proportional limit, showing that under a precise relation between the scaling parameters of the shortcut and residual branches, the feature covariance converges to the solution of a weighted version of the neural covariance SDE for MLPs (Theorem 3.2). The dependence on the depth-to-width ratio implies the existence of a stable non-commutative limit for residual networks, complementing the commutative limit studied in Hayou and Yang [50].

2. We propose *shaped attention*, where we modify the Softmax-based attention mechanism to be a perturbation of the identity. We demonstrate that shaped attention successfully prevents the degeneracy of correlation in contrast to existing Transformer architectures (Figure 1). The enhanced stability in the forward pass is reflected in the gradients, which are also stable with depth, as we empirically show in Figure 2.

3. For the proposed shaped attention architecture, we derive the neural covariance SDE characterizing the initial distribution in the proportional limit (Theorem 4.2). Consequently, we provide the first characterization of Transformer-type architectures, i.e. the shaped Transformer, in the large depth-and-width regime (Corollary 4.3).

4. We provide simulations to validate the theory and to interpret the effects of network hyperparamaters on the covariance matrix of the shaped Transformer. Specifically, we study finite time stability of the SDE and provide explicit guidance on hyperparameters to prevent numerical instability.

The paper is organized as follows: In Section 2, we provide the basic setup and some background on existing results. In Section 3, we generalize the SDE results of M. Li et al. [18] to include skip connections. This serves as a model to understand the effect of skip connections in isolation from the attention model. In Section 4, we present our main result, first pinpointing the origins of instability in the Softmax, then showing how the modifications underlying *shaped attention* allow us to derive a non-trivial SDE limit. Finally, in Section 5, we discuss the implications of our results and some future directions. Proofs of all theorems and additional experiments are deferred to the Appendix.

## 2   Background

**Setup.** Let $X_\ell \in \mathbb{R}^{m \times n}$ be the data matrix representing a sequence of $m$ tokens embedded in $n$ dimensions at layer $\ell \in [d]$, where $d$ is the depth of the network. We elide the explicit dependence on $\ell$ when it is clear from the context, and use superscript Greek letters to indicate specific tokens' representations, for instance $x_\ell^\alpha \in \mathbb{R}^n$ is the $\alpha$-th row of $X_\ell$. We consider the following attention model with residual connections:

$$X_{\ell+1} = \lambda X_\ell + \gamma A_\ell X_\ell \frac{1}{\sqrt{n}} W_\ell^V \tag{1}$$

where $\gamma, \lambda \in [0, 1]$ are parameters that control the strength of the shortcut and residual branch, respectively, $W_\ell^V \in \mathbb{R}^{n \times n}$ is the weight matrix of the values, and $A_\ell \in \mathbb{R}^{m \times m}$ is the attention matrix. We consider Softmax-based scaled dot-product attention, where $A_\ell$ has the form:

$$A_\ell = \text{Softmax} \left( \frac{1}{\tau} X_\ell \frac{1}{\sqrt{n}} W_\ell^Q \frac{1}{\sqrt{n}} W_\ell^{K,\top} X_\ell^\top \right), \tag{2}$$

where the Softmax is applied row-wise, $W_\ell^Q, W_\ell^K \in \mathbb{R}^{n \times n_k}$ are additional random weights, and $\tau$ is a temperature parameter, which controls the entropy of the distribution. Here we let all the weight matrices $W_\ell^Q, W_\ell^K, W_\ell^V$ have $\mathcal{N}(0, 1)$-iid entries. In the case where $\lambda, \gamma = 1$, with the application of LayerNorm on the residual branch [48], and with $\tau = \sqrt{n_k}$, we recover the attention block of the vanilla "Pre-LN" Transformer architecture [2]. Here we note that we pull the conventional $n^{-1/2}$ factor outside of the weight matrices, which preserves the forward pass, and yields equivalent training dynamics up to a reparameterization of the learning rate [25]. In this work, we consider unnormalized architectures, and control the variance propagation with the condition $\lambda^2 + \gamma^2 = 1$ [40]. We are interested in studying the so-called *neural covariance* for the attention model (Eq. 1) in the proportional limit.

**Neural Covariance.** In deep learning theory, researchers have long sought to understand how networks internally represent different inputs and how different architectural choices affect these representations. The approach followed by work on signal propagation has been to study how the relative alignment of different inputs evolves across the network, as measured by the neural covariance $V_\ell^{\alpha\beta} := \frac{1}{n} \langle x_\ell^\alpha, x_\ell^\beta \rangle$ (or $\rho^{\alpha\beta} := (V_\ell^{\alpha\alpha} V_\ell^{\beta\beta})^{-1/2} V_\ell^{\alpha\beta}$ if interested only in the correlation). At initialization, characterizations of this covariance structure have been exploited to infer important properties of neural networks [10, 11]. As an example, in the sequential infinite-width-*then*-depth limit, the correlation $\rho_d^{\alpha\beta}$ of MLPs is known to converge to a fixed point independent of the input [11, 16]. In this regime, the model is not able to discriminate different data points, which severely hinders training, as the gradient step for the deep layers is taken in the same direction regardless of the input. In the context of Softmax-based attention models, Dong et al. [8] proved that the feature matrix $X_\ell$ loses rank doubly exponentially fast with depth, and Noci et al. [9] showed how this leads to vanishing gradients of the queries and keys parameters, thus further highlighting how the stability of forward and backward passes are deeply entangled (see also Figure 2).

**Stabilizing the Effect of Non-Linear Layers.** Central to the issue of degeneracy of the neural covariance are commonly used non-linear activation functions that severely deviate from the identity. The recent line of work of Deep Kernel Shaping (DKS) [16–18] addresses the issue by considering the cumulative amount of non-linearity throughout layers, and *shaping* the activation function by making it closer to the identity map. Inspired by this line of work, B. He et al. [46] devise an initialization for Transformers that avoid the rank collapse problem without the aid of skip connections or LayerNorm.

In an alternative approach, the line of work behind Stable ResNets [50–53] considers scaling the residual branches by $\gamma = 1/\sqrt{\text{depth}}$, and postulates this scaling is sufficient to stabilize the neural

covariance with minimal assumptions on the activation function. Noci et al. [9] adopts this scaling to give precise formulas on the expected covariance of a Transformer at initialization. In this work, we consider $\gamma$ constant in width and depth, and derive a complementary limiting result.

**The Proportional Infinite-Depth-and-Width Limit.** In the context of feed-forward MLPs, the output distribution with respect to a single input was studied in [37, 41], where it was shown that for the ReLU nonlinearity, the norm of the activations $V^{\alpha\alpha}$ converges to a log-normal random variable. To resolve the degeneracy of covariance and provide a characterization of output distributions for *multiple inputs*, M. Li et al. [18] shapes the ReLU by setting its slope $1/\sqrt{\text{width}}$-away from linearity. In the proportional limit, the effect of the non-linearity accumulates over the $d$ layers, and the covariance matrix $V_\ell = [V_\ell^{\alpha\beta}]_{\alpha\beta}$ converges weakly to the solution of the SDE

$$dV_t = b_{\text{ReLU}}(V_t)\,dt + \Sigma_{\text{lin}}^{1/2}(V_t)\,dB_t\,, \tag{3}$$

where the formulae for coefficients $b_{\text{ReLU}}, \Sigma_{\text{lin}}$ can be found in Theorem 3.2.

We note that the output neuron distributions are directly recovered as a conditional Gaussian with covariance $V_T$ for $T = \frac{d}{n}$, in a similar spirit as the neural network Gaussian process (NNGP) results [20–22]. For example, the $i$-th output $X_{\text{out},i}$ conditioned on $V_d$ are asymptotically iid $\mathcal{N}(0, V_T)$ as $d, n \to \infty$. The reader is referred to Appendix A for more technical background on the covariance SDE and the convergence result.

While the existing results are limited to initialization, we remind the reader that this is a necessary step before we can study training dynamics. In particular, the NNGP techniques developed for infinite-width networks at initialization were directly used to study the training dynamics in the same limit [24, 54]. We will provide further discussions on this topic in Section 5.

# 3 Warm-Up: a Neural Covariance SDE for ResNets

To understand the effect of skip connections, it is helpful to look at a simplified model composed of a shaped ReLU-activated layer and skip connections:

$$X_{\ell+1} = \lambda X_\ell + \gamma \sigma_s\left(X_\ell \frac{1}{\sqrt{n}} W_\ell^{\text{pre}}\right)\sqrt{\frac{c}{n}} W_\ell^{\text{post}}\,, \tag{4}$$

where $\sigma_s(x) := s_+ \max(x, 0) + s_- \min(x, 0)$ is the shaped ReLU with slopes $s_\pm := 1 + c_\pm n^{-1/2}$ for some constants $c_+, c_- \in \mathbb{R}$ . We assume i.i.d weights $(W_\ell^{\text{pre}})_{ij}, (W_\ell^{\text{post}})_{ij} \overset{\text{iid}}{\sim} \mathcal{N}(0, 1)$, and $c^{-1} = \mathbb{E}\,\sigma_s(g)^2$ for $g \sim N(0, 1)$ is a constant that ensures that the activations are normalized [55]. Notice that this is the form of the feedforward layer in a Transformer [2].

We will next define the notion of convergence for our covariance matrices and state our first main result. We refer the reader to Appendix A for more precise details on the Skorohod topology.

**Definition 3.1** (Convergence of Covariance). *Let $X_\ell \in \mathbb{R}^{m \times n}$ be the $\ell$-th layer matrix of representations, and define the feature covariance as $V_\ell = \frac{1}{n} X_\ell X_\ell^\top$. Let $V_t^{(n)} = V_{\lfloor tn \rfloor} \in \mathbb{R}^{m(m+1)/2}$ be the the continuous time interpolation of the upper triangular entries as a vector. We say the covariance $V^{(n)}$ converges to $V$, if in the limit as $n, d \to \infty$, $\frac{d}{n} \to T$, the process $\{V_t^{(n)}\}_{t \in [0,T]}$ converges to $\{V_t\}_{t \in [0,T]}$ weakly in the Skorohod topology.*

**Theorem 3.2.** *Let $X_\ell$ be the hidden layers of a ResNet defined in Eq. 4 with $\lambda^2 + \gamma^2 = 1$, where both $\lambda$ and $\gamma$ do not depend on $d, n$. Then the feature covariance $V_\ell$ converges to the solution of the following SDE (in the sense of Definition 3.1)*

$$dV_t = b_{res}(V_t)\,dt + \Sigma_{res}(V_t)^{1/2}\,dB_t\,, \quad V_0 = \frac{1}{n} X_0 X_0^\top\,, \tag{5}$$

*where $b_{res}(V) = \gamma^2 b_{ReLU}(V) = \gamma^2[\nu(\rho^{\alpha\beta})\sqrt{V^{\alpha\alpha}V^{\beta\beta}}]_{\alpha \leq \beta}$ with $\rho^{\alpha\beta} = V^{\alpha\beta}(V^{\alpha\alpha}V^{\beta\beta})^{-1/2}$ and*

$$\nu(\rho) = \frac{(c_+ - c_-)^2}{2\pi}\left(\sqrt{1 - \rho^2} - \rho \arccos \rho\right)\,, \tag{6}$$

*furthermore, $\Sigma_{res}(V) = 2\gamma^2 \Sigma_{lin}(V) = 2\gamma^2[V^{\alpha\delta}V^{\beta\omega} + V^{\alpha\omega}V^{\beta\delta}]_{\alpha \leq \beta, \delta \leq \omega}$.*

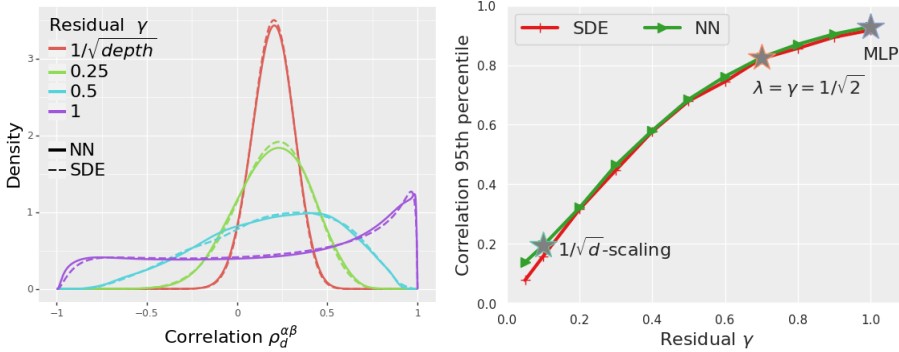

Figure 3: Left: Kernel density estimates of correlation $\rho_d^{\alpha\beta}$ for various values of the residual strength parameter $\gamma$. In particular, $\gamma = 1$ recovers a shaped-ReLU MLP without skip connections, and $\gamma = 1/\sqrt{d}$ is the setting studied in Noci et al. [9] and Hayou and Yang [50]. Solid lines represent finite networks, while our SDE simulations are presented in dashed lines. Right: 95th percentile of the absolute value of the correlation distribution as a function of $\gamma$. Note reducing $\gamma$ reduces the concentration around $\rho^{\alpha\beta} = 1$, and our SDE reliably approximates finite size networks. Simulated with $n = 300, d = 100, \rho_0^{\alpha\beta} = 0.2, c_+ = 0, c_- = -1$, and $2^{13}$ samples.

Notice how the limiting SDE closely resembles the MLP case (Eq. 3), which is recovered exactly when $\gamma = 1$. The only difference is the extra 2 factor, which comes from the fact that in our definition each layer has effectively two times the number of weight matrices than the standard formulation for MLPs. As the drift depends solely on the nonlinearity, and the diffusion depends soley on the random weights, only the diffusion variance is doubled. The residual branch parameter $\gamma < 1$ dampens both the drift and the variance of the Brownian motion by $\gamma^2$, thus it can be interpreted as a time change. In other words, the effect of $\gamma$ at initialization is equivalent to reducing depth-to-width ratio, inline with existing intuitions that ResNets have a lower "effective-depth" [56]. To visualize the stabilizing effect of $\gamma$ on the distribution, in Figure 3 (right) we plot the 95th percentile correlation as a function of $\gamma$. The increasing trend indicates a larger probability of perfect alignment between two tokens. In Figure 3 (left) we plot the densities of both the residual SDE and the corresponding residual network for various values of $\gamma$. Notice how the samples from the SDE well-approximates the histogram of a finite network.

**A Stable Non-Commutative Limit.** Our results complement those of Hayou and Yang [50], where the authors have shown that for a similar ResNet under the parameter scaling $\lambda = 1, \gamma = 1/\sqrt{d}$, the depth and width limits *commute*. More precisely, the covariance $V^{\alpha\beta}$ converges to the same limit regardless of the order with respect to which the limit is taken or the depth-to-width ratio. Furthermore, the limit is *deterministic*, and can be described by an ordinary differential equation (ODE). Intuitively, the convergence to a deterministic quantity occurs because $\gamma = 1/\sqrt{d}$ suppresses the random fluctuations enough to vanish in the limit. On the other hand, our results show that for $\lambda, \gamma$ constant in $n, d$, the random fluctuations are on the right order of $O(n^{-1/2})$ as in the MLP case (Eq. 3), hence they do not vanish in the simultaneous limit. The most notable difference is that our limiting regime is *non-commutative* as it depends on the depth-to-width ratio of the network. We remark that both regimes prevents degeneracy of covariance for residual architectures, forming two theories that complement each other.

## 4 Neural Covariance SDE for Softmax-Based Attention

### 4.1 Unshaped Attention and Its Taylor Expansion

A central piece to the neural covariance SDE theory for MLPs [18] is identifying the exact scaling of shaped activation functions. In particular, the effect of the activations on the covariance Markov chain $V_\ell$ must be on the same order as the random weights in an MLP, thus forming an approximate

Euler-discretization

$$V_{\ell+1} = V_\ell + \frac{b(V_\ell)}{n} + \frac{\Sigma(V_\ell)^{1/2}\xi_\ell}{\sqrt{n}} + O(n^{-3/2}), \tag{7}$$

where $b, \Sigma$ are deterministic coefficients, and $\xi_\ell$ are random vectors with zero mean and identity covariance. From here onwards, we use $O(n^{-p})$ to denote a random variable $Z$ such that $n^p Z$ has all moments bounded by universal constants (i.e. independent of $n$). Since the update can be interpreted as discretization with step size $n^{-1}$, naturally the Markov chain converges to an SDE. We again note that a stable SDE implies a stable covariance structure for finite size networks.

To achieve the same goal for modified attention mechanisms, we consider a similar approach as M. Li et al. [18] for smooth activation functions, and Taylor expand the Softmax function in terms of a large temperature parameter $\tau$. To this end, let $Y_\ell$ to be the matrix of dot-products between queries, and keys, i.e. $Y_\ell := X_\ell \frac{1}{\sqrt{n}} W_\ell^Q \frac{1}{\sqrt{n}} W_\ell^{K,\top} X_\ell^\top$.

More specifically, given a row $y^\alpha \in \mathbb{R}^{1 \times m}$ of the logits $Y_\ell \in \mathbb{R}^{m \times m}$, we can Taylor expand the row-wise Softmax in terms of $\tau^{-1}$:

$$\text{Softmax}(\tau^{-1}y^\alpha) = \frac{1}{m}\mathbf{1}^\top + \frac{1}{\tau m}(y^\alpha - \overline{y^\alpha}) + \frac{1}{2\tau^2 m}\left[(y^\alpha - \overline{y^\alpha})^2 - \left(\overline{y^{\alpha^2}} - \overline{(y^\alpha)^2}\right)\right] + O(\tau^{-3}), \tag{8}$$

where $\overline{y^\alpha} := \frac{1}{m}\sum_\beta y^{\alpha\beta}\mathbf{1}^\top$ and $(y^\alpha)^2$ is the vector with squared entries of $y^\alpha$, and $\mathbf{1} \in \mathbb{R}^{m \times 1}$ is the (column) vector of ones. We note in practice $\tau$ is often set to $\sqrt{n_k}$, which is often quite large and allows for asymptotic analysis [9].

We observe that the zero-th order term $m^{-1}\mathbf{1}^\top$ is independent of $\tau$. Considering the form of the attention block as $A_\ell X_\ell \frac{1}{\sqrt{n}} W_\ell^V$, this yields an update that is no longer a small perturbation of $V_\ell$, regardless of how $\tau$ is chosen. Therefore, to form a Markov chain like Eq. 7, we actually require $A_\ell$ to be approximately the identity matrix.

## 4.2 Shaped Attention

To shape the Softmax-attention mechanism as a perturbation of the identity matrix, we propose the following modifications which we call the *shaped attention* [5]

$$A_\ell = I + \text{Softmax}(\tau^{-1}Y_\ell) - \frac{1}{m}\mathbf{1}\mathbf{1}^\top, \quad \tau = \tau_0\sqrt{nn_k}. \tag{9}$$

The shaped attention presents three modifications to the Softmax attention in Eq. 2. Firstly, the zero-order term $m^{-1}\mathbf{1}\mathbf{1}^\top$ of the Taylor expansion (Eq. 8) is removed as it causes a non-infinitesimal drift in the Markov Chain that ultimately leads to instability in the covariance (see Section 4.1). Secondly, we also observe that when $\tau$ is very large, the centered Softmax is a perturbation around zero. To recover an approximate Euler-update like in Eq. 7, we simply add back the identity matrix. By biasing the attention matrix towards the identity, we encourage each token to self-attend. This type of modification was also previously considered by B. He et al. [46]. Finally, the Softmax's temperature is chosen to scale as $\tau = \tau_0\sqrt{nn_k}$, for some constant $\tau_0 > 0$, which guarantees a non-degenerate limit as $(d, n) \to \infty$ (Theorem 4.2). Note that the extra $\sqrt{n}$ term is a departure from the standard parameterization.

In Figure 4, we show how removing any of the proposed changes individually alters the neural covariance structure, which becomes degenerate for large depths, while the proposed modifications remain stable. We stress that here for simplicity we focus on attention without masking. Shaped attention can be extended to include masking (e.g. casual masking) by centering each i-th row of the Softmax matrix by a different factor $1/m_i$, where $m_i$ is the number of un-masked tokens in the i-th row.

---

[5]In principle, it could be possible to have a close-to-identity Softmax matrix when the logits are large. However, this regime also corresponds to a very saturated Softmax, thus making training unstable [57]. As a result, we will avoid this direction in this work.

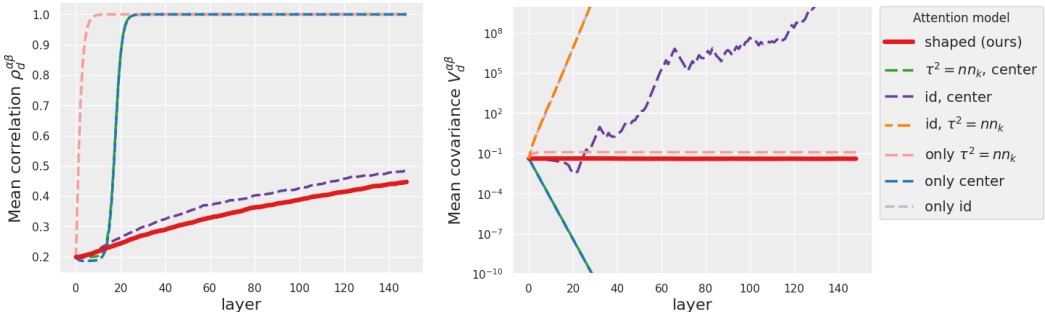

Figure 4: Mean correlation (left) and covariance (right) (in absolute value) under various interventions on the proposed shaped attention. In particular, we remove either one or two of the three modifications from the shaped attention in Eq. 9. For instance "$\tau^2 = nn_k$, center" indicates that we use the proposed temperature, and we center by $m^{-1}$, but we do not add the identity matrix, while in "only id" we add the identity matrix but use $\tau = \sqrt{n_k}$ and do not center. We note in this "only id" case, the covariance remains unstable due to incorrect scaling. Due to exploding covariance, we choose to not include the cases "id, $\tau^2 = nn_k$" and "only id" in the correlation plot (but only in the covariance plot). Simulated with $n = 300, d = 150, \rho_0^{\alpha\beta} = 0.2, \gamma = 1/\sqrt{2}$ and $2^{13}$ samples.

### 4.3 Main Result – Neural Covariance SDEs for Shaped Attention Models and Shaped Transformers

Before we state our main results, we will first define a weakened notion of convergence, which is required whenever the drift and covariance coefficients are not Lipschitz. This was also required for the case of shaped MLPs with smooth activations [18].

**Definition 4.1** (Local Convergence). *We say the covariance $V^{(n)}$ converges locally to $V$ if the stopped process $\{V_{t \wedge T_r}^{(n)}\}_{t \geq 0}$ converges to $\{V_{t \wedge T_r}\}_{t \geq 0}$ in the sense of Definition 3.1 for all stopping times of the form $T_r = \inf\{t > 0 : \|V_t\| \geq r\}$ with $r > 0$.*

Let the covariance with respect to the average token be defined as $V^{\alpha\bar{x}} := m^{-1}\sum_{\nu=1}^m V^{\alpha\nu}$, and the average trace be $\bar{V} := m^{-1}\sum_{\nu=1}^m V^{\nu\nu}$. We will need to compute a couple of important moments from the Taylor expansion terms of the Softmax (Lemma C.2)

$$
\begin{aligned}
S_1^{\alpha\delta,\beta\omega} &:= n_k^{-1}\mathbb{E}(Y^{\alpha\delta} - \overline{y^\alpha})(Y^{\beta\omega} - \overline{y^\beta}) = V^{\alpha\beta}\left(V^{\delta\omega} - V^{\delta\bar{x}} - V^{\omega\bar{x}} + V^{\bar{x}\bar{x}}\right), \\
S_2^{\alpha\delta} &:= n_k^{-1}\mathbb{E}\left[(Y^{\alpha\delta} - \overline{y}^\alpha)^2 - (\overline{(Y^\alpha)^2} - \overline{y^\alpha}^2)\right] = V^{\alpha\alpha}\left(V^{\delta\delta} - 2V^{\delta\bar{x}} + 2V^{\bar{x}\bar{x}} - \bar{V}\right).
\end{aligned} \tag{10}
$$

We are now ready to state our main result.

**Theorem 4.2.** *Let $X_\ell$ be the hidden layers of a residual attention network defined in Eq. 1 with shaped attention in Eq. 9, parameters $\lambda^2 + \gamma^2 = 1$ and $\tau = \tau_0\sqrt{nn_k}$, where $\lambda, \gamma, \tau_0$ all do not depend on $d, n$. Then the feature covariance $V_\ell$ converges locally to the solution of the following SDE (in the sense of Definition 4.1)*

$$
dV_t = b(V_t)dt + \Sigma(V_t)^{1/2}dB_t, \quad V_0 = \frac{1}{n}X_0 X_0^\top,
$$

*where the drift has the following form*

$$
b(V) = \frac{\gamma^2}{\tau_0^2}\left[\frac{1}{m^2}\sum_{\nu,\kappa=1}^m V^{\nu\kappa}S_1^{\alpha\nu,\beta\kappa} + \frac{1}{2m}\sum_{\nu=1}^m(V^{\beta\nu}S_2^{\alpha\nu} + V^{\alpha\nu}S_2^{\beta\nu})\right]_{\alpha \leq \beta},
$$

*the diffusion coefficient is defined by $\Sigma(V) = \gamma^2(2 - \gamma^2)\Sigma_{lin}(V) + \gamma^4\tau_0^{-2}[\mathcal{A}^{\alpha\beta,\delta\omega}]_{\alpha \leq \beta, \delta \leq \omega}$, and*

$$
\mathcal{A}^{\alpha\beta,\delta\omega} := \frac{1}{m^2}\sum_{\nu,\kappa=1}^m\left(V^{\alpha\kappa}V^{\delta\nu}S_1^{\beta\kappa,\omega\nu} + V^{\alpha\kappa}V^{\omega\nu}S_1^{\beta\kappa,\delta\nu} + V^{\beta\nu}V^{\delta\kappa}S_1^{\alpha\nu,\omega\kappa} + V^{\beta\nu}V^{\omega\kappa}S_1^{\alpha\nu,\delta\kappa}\right).
$$

The drift depends on the shaped attention mechanism through $S_1^{\alpha\delta,\beta\omega}$ and $S_2^{\alpha\delta}$, the moments of the first and second order terms of the Softmax's Taylor expansion. On the other hand, the diffusion term depends on the attention solely through $S_1$, present in the additional term $\mathcal{A}^{\alpha\beta,\delta\omega}$. The presence of $\mathcal{A}^{\alpha\beta,\delta\omega}$ is an intriguing difference compared to shaped ReLU networks, where the diffusion is not affected by the activation function. Both components of the SDE depend on averages over the tokens, reflecting the mixing property of the self-attention mechanism, in which every pair of tokens is compared through dot products to form the attention weights. Finally, notice how the residual branch parameter $\gamma^2$ has a dampening effect on the scale of both the drift and the diffusion in a similar way as in fully-connected residual network.

We are now ready to introduce the full shaped Transformer architecture, where we combine the attention and residual layers:

$$Z_\ell = \lambda X_\ell + \gamma A_\ell X_\ell \frac{1}{\sqrt{n}} W_\ell^V, \quad X_{\ell+1} = \lambda Z_\ell + \gamma \sigma_s \left( Z_\ell \frac{1}{\sqrt{n}} W_\ell^{\mathrm{pre}} \right) \sqrt{\frac{c}{n}} W_\ell^{\mathrm{post}}, \quad (11)$$

where $A_\ell$ is the shaped attention defined by Eq. 9. We note that covariance SDE handle stacking of different layer types very conveniently by simply adding the drift and covariance of the diffusion coefficients, which we summarize in the Corollary below.

**Corollary 4.3** (Shaped Transformer Covariance SDE). *Let $X_\ell$ be the hidden layers of a shaped transformer defined in Eq. 11 with parameters $\lambda^2 + \gamma^2 = 1$ and $\tau = \tau_0 \sqrt{n n_k}$, where $\lambda, \gamma, \tau_0$ all do not depend on $d, n$. Then the feature covariance $V_\ell$ converges locally to the solution of the following SDE (in the sense of Definition 4.1)*

$$dV_t = [b(V_t) + b_{res}(V_t)] \, dt + [\Sigma(V_t) + \Sigma_{res}(V_t)]^{1/2} \, dB_t, \quad (12)$$

*where the coefficients are defined in Theorem 3.2 and Theorem 4.2.*

### 4.4 On Finite Time Stability of the SDE and Shaped Attention Networks

Although we did not observe numerical instability in majority of our simulations of the shaped attention networks and the corresponding SDE, we did observe that the drift component $b(V_t)$ in Theorem 4.2 is cubic in the entries of $V_t$. Whenever the drift is not Lipschitz as in this case, we do not have general guarantees for the existence of a solution for all time (see the Feller test for explosions [58, Theorem 5.5.29]). In fact, MLPs with smooth activations also yield non-Lipschitz drift coefficients as seen in M. Li et al. [18].

However, locally Lipschitz coefficients are sufficient to guarantee the existence of local solutions, in the sense of up to a stopping time [59, Proposition 6.9]. Not only does this fact help us establish a precise notion of convergence (Definition 4.1), we can also study the practical implications of this for finite sized attention networks. More specifically, we can inspect the effect of architectural changes to a stopping time.

To demonstrate the potential numerical instabilities, we had to choose an *adversarial* set of parameters: in particular, an unrealistically large norm (approx. $10\sqrt{n}$) for the initial tokens $X_0$, which enlarges the eigenvalues of $V_0$ to the order of $100$. Given these initial conditions and a large residual connection weight $\gamma$, we were able to consistently generate numerically unstable behaviour in shaped attention networks (see Figure 5 (left)).

That being said, it is very straight forward to stabilize the network by tweaking parameters such as $\gamma, \tau_0$ and the depth-to-width ratio of the network. We demonstrate the effect of tuning $\gamma$ on both sample trajectories of the maximum eigenvalue of $V_\ell$ and the stopping time in Figure 5. As we may intuitively expect, tuning $\gamma$ smaller will delay the time scale of numerical instabilities, hence allowing for larger depth networks to remain stable.

## 5 Discussion

**Architecture Design and Hyperparameter Tuning.** Previous work have demonstrated the practical impact scaling limits can have on designing activation functions [16, 17] and tuning hyperparameters [19]. We follow this line of motivations and proposed a novel attention mechanism, which successfully stabilizes the covariance structure in arbitrarily deep Transformers (e.g. Figure 1). The natural next

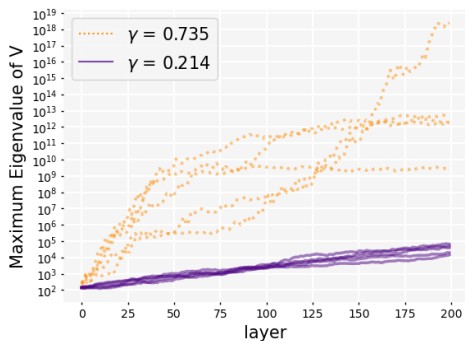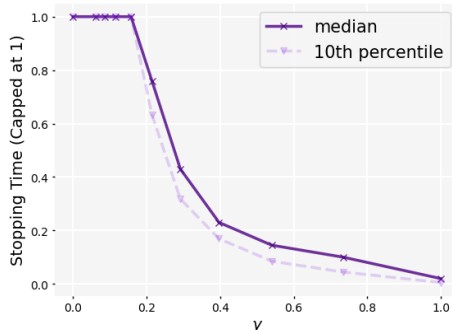

Figure 5: Left: Trajectories of the maximum eigenvalue of the covariance matrix in a shaped attention network, with *adversarially* large initial condition. Right: Stopping time of the shaped attention neural network, capped at 1. Stopping time is defined as $t^* = d^*/n$ with $d^*$ the maximum depth beyond which one of the eigenvalues of the covariance matrix exceeds $10^4$ or drops below $10^{-4}$. Simulated with $n = d = 200$, $\tau_0 = 1$, and 100 samples used for median and 10th percentile.

step is to investigate the scaling of gradients in the infinite-depth-and-width limit. As Yang et al. [19] illustrated, the existence of an infinite-width limit for the gradient implies the optimal hyperparameters for the training algorithm will also converge. This type of results allows for tuning of hyperparameters on networks with a much smaller width, yet extends easily to arbitrarily large networks that approximates the same limit, saving massive amounts of computing cost in the process. Given the existence of an infinite-depth-and-width limit for the forward pass, we believe it's possible to extract optimal hyperparameters from networks with not only a much smaller width, but *smaller depth* as well.

**Preliminary Experiments.** Although this work is primarily theoretical, it is important to consider whether or not the proposed architecture is useful in practice. Given limited computing resources, we chose to only briefly test the feasibility of training the shaped Transformer. Nevertheless, our preliminary experiments show promising results when it comes to training stability. In particular, the shaped Transformer (without LayerNorm) does indeed train at approximately the same speed as well tuned Transformer architectures. Full details of the experiment and results can be found in Appendix D. A more comprehensive set of experiments with different tasks, datasets, and larger networks will be required to confidently determine the practical feasibility of the shaped Transformer, which we defer to future work.

**Training Dynamics and Generalization.** As mentioned in the introduction, the limitations of infinite-width NTK theories motivates our study of the proportional infinite-depth-and-width limit. In particular, to address many of the open problems in deep learning theory, we need a faithful and tractable description of training dynamics. Given the results at initialization, the proportional limit holds the potential for such a theory of training as well. Another promising indicator is that deep networks learn features in the proportional regime [38], which has been identified as a key advantage of neural networks over kernel methods [25, 60–66]. A precise theory of training will help us understand other types of instabilities during training and improve existing optimization methods. Furthermore, determining the network which training converges to is a necessary step towards a theory of generalization, as demonstrated by the infinite-width approach [67]. In light of our results, we believe that our theory sets the stage for future work on training and generalization in deep learning.

# Acknowledgement

CL and ML would like to thank Keiran Paster for insightful discussions. LN would like to thank Sotiris Anagnostidis for support in pre-processing the dataset used for the training experiments of this manuscript. ML is supported by the Ontario Graduate Scholarship and Vector Institute. DMR is supported in part by Canada CIFAR AI Chair funding through the Vector Institute, an NSERC Discovery Grant, Ontario Early Researcher Award, a stipend provided by the Charles Simonyi Endowment, and a New Frontiers in Research Exploration Grant.

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

# A Preliminaries: Covariance SDE Framework

In this section, we will review existing results on Markov chain convergence to an SDE, as well as existing results for the covariance SDE. See also the Appendix of M. Li et al. [18] for more details.

Firstly, we will define the Skorohod $J_1$-topology, or just the Skorohod topology for short [68, Appendix 5]. The topology is used to describe convergence of continuous time processes with discontinuities, in particular, Markov chains with a continuous time interpolation fits in this category. Let $S$ be a complete separable metric space, and $D_{\mathbb{R}_+, S}$ be the space of càdlàg functions (right continuous with left limits) from $\mathbb{R}_+ \to S$. We use $x_n \xrightarrow{ul} x$ to denote locally uniform convergence (uniform on compact subsets of $\mathbb{R}_+$), and consider the class of bijections $\lambda$ on $\mathbb{R}_+$ so that $\lambda$ is strictly increasing with $\lambda_0 = 0$ (can be interpreted as a time change).

**Definition A.1.** *We define **Skorohod convergence** $x_n \xrightarrow{s} x$ on $D_{\mathbb{R}_+, S}$ if there exists a sequence of bijections $\lambda_n$ satisfying above conditions and*

$$\lambda_n \xrightarrow{ul} Id, \quad x_n \circ \lambda_n \xrightarrow{ul} x. \tag{13}$$

*In particular, we call the topology that induces this convergence the **Skorohod topology** [68, Theorem A5.3].*

On a heuristic level, if we have sequences of Markov chains $Y^n$ that satisfy the following type of Euler updates

$$Y_{\ell+1}^n = Y_\ell^n + \frac{b(Y_\ell^n)}{n} + \frac{\sigma(Y_\ell^n)\xi_\ell}{\sqrt{n}}, \tag{14}$$

where $\xi_\ell$ are iid random variables with zero mean and identity covariance, then we can interpolate this process in continuous time with $X_t^n = Y_{\lfloor tn \rfloor}^n$ and show that as $n \to \infty$, we have that $X^n$ converges to the solution of the following SDE (weakly with respect to the Skorohod topology)

$$dX_t = b(X_t)\,dt + \sigma(X_t)\,dB_t, \quad X_0 = \lim_{n \to \infty} Y_0^n. \tag{15}$$

Our next theorem essentially weakens this result in several ways. Firstly, we don't need to take the step size $n^{-1}$, but instead replace it with $n^{-2p}$ for all $p > 0$. Next, we can allow the update to contain higher order terms that vanish, in particular, any terms of order $O(n^{-2p-\delta})$ for all $\delta > 0$. Here, we remind the readers that $O(n^{-p})$ denotes a random variable $Z$ such that $n^p Z$ has all moment bounded by a universal constant (independent of $n$). Thirdly, we can allow $b(y)$ to be random, or more precisely replace it with $\widehat{b}(y, \omega_n)$ such that $\mathbb{E}\widehat{b}(y, \omega_n) = b(y)$. Fourthly, we can also allow $b, \sigma$ to be a sequence $b_n, \sigma_n$ such that they converge to $b, \sigma$ on uniformly on compact sets. Finally, we can also weaken the topology of convergence to locally, that is all processes are stopped by a stopping time as in Definition 4.1.

Now we will state the main technical result in this section.

**Proposition A.2** (Convergence of Markov Chains to SDE, Proposition A.6, [18]). *Let $Y^n$ be a discrete time Markov chain on $\mathbb{R}^N$ defined by the following update for $p, \delta > 0$*

$$Y_{\ell+1}^n = Y_\ell^n + \frac{\widehat{b}_n(Y_\ell^n, \omega_\ell^n)}{n^{2p}} + \frac{\sigma_n(Y_\ell^n)}{n^p}\xi_\ell^n + O(n^{-2p-\delta}), \tag{16}$$

*where $\xi_\ell^n \in \mathbb{R}^N$ are iid random variables with zero mean, identity covariance, and moments uniformly bounded in $n$. Furthermore, $\omega_\ell^n$ are also iid random variables such that $\mathbb{E}[\widehat{b}_n(Y_\ell^n, \omega_\ell^n)|Y_\ell^n = y] = b_n(y)$ and $\widehat{b}_n(y, \omega_\ell^n)$ has uniformly bounded moments in $n$. Finally, $\sigma_n$ is a deterministic function, and the remainder terms in $O(n^{-2p-\delta})$ have uniformly bounded moments in $n$.*

*Suppose $b_n, \sigma_n$ are uniformly Lipschitz functions in $n$ and converges to $b, \sigma$ uniformly on compact sets, then in the limit as $n \to \infty$, the process $X_t^n = Y_{\lfloor tn^{2p} \rfloor}^n$ converges in distribution to the solution of the following SDE in the Skorohod topology of $D_{\mathbb{R}_+, \mathbb{R}^N}$*

$$dX_t = b(X_t)\,dt + \sigma(X_t)\,dB_t, \quad X_0 = \lim_{n \to \infty} Y_0^n. \tag{17}$$

*Suppose otherwise $b_n, \sigma_n$ are only locally Lipschitz (but still uniform in $n$), then $X^n$ converges locally to $X$ in the same topology (see Definition 4.1). More precisely, for any fixed $r > 0$, we consider the stopping times*

$$\tau^n := \inf\{t \geq 0 : |X_t^n| \geq r\}, \quad \tau := \inf\{t \geq 0 : |X_t| \geq r\}, \tag{18}$$

*then the stopped process $X_{t \wedge \tau^n}^n$ converges in distribution to the stopped solution $X_{t \wedge \tau}$ of the above SDE in the same topology.*

We will briefly recall the main result of M. Li et al. [18] next. As mentioned earlier in the text, the setting is for MLPs defined as follows

$$X_{\ell+1} = \sigma_s(X_\ell)\sqrt{\frac{c}{n}}W_\ell, \tag{19}$$

where $\sigma_s(x) = s_+ \max(x, 0) + s_- \min(x, 0)$ with $s_\pm = 1 + \frac{c_\pm}{\sqrt{n}}$, $c^{-1} = \mathbb{E}\,\sigma_s(g)^2$ for $g \sim \mathcal{N}(0,1)$, $n$ is the width of the network, and $W_{\ell,ij} \sim \mathcal{N}(0,1)$ are iid random weights.

Then it was shown that in the limit as $n, d \to \infty$ with $\frac{d}{n} = T > 0$, the covariance matrix $V_\ell = \frac{1}{n}X_\ell X_\ell^\top$ (or equivalent the post activation $\frac{c}{n}\sigma_s(X_\ell)\sigma_s(X_\ell)^\top$ since the activation becomes infinitesimal) converges the solution of the following SDE in the Skorohod topology [18, Theorem 3.2]

$$dV_t = b_{\text{ReLU}}(V_t)\,dt + \Sigma_{\text{lin}}(V_t)^{1/2}\,dB_t, \quad V_0 = \frac{1}{n}X_0 X_0^\top, \tag{20}$$

where the coefficients were given in Theorem 3.2.

We remark that if the output is defined as $X_{\text{out}} = \frac{1}{\sqrt{n}}W_{\text{out}}X_d \in \mathbb{R}^{n_{\text{out}}}$, then we can recover the output distribution as

$$X_{\text{out}} \sim N(0, V_T \otimes I_{n_{\text{out}}}), \tag{21}$$

where we treated $X_{\text{out}}$ as a vector in $\mathbb{R}^{mn_{\text{out}}}$.

# B SDE for Residual Network

Recall that we adopt the following model:

$$X_{\ell+1} = \lambda X_\ell + \gamma\frac{1}{\sqrt{n}}\sigma_s\left(\sqrt{\frac{c}{n}}X_\ell W_\ell^{\text{pre}}\right)W_\ell^{\text{post}}, \tag{22}$$

where $\sigma_s(x) := s_+ \max(x, 0) + s_- \min(x, 0)$ is the shaped ReLU with slopes $s_\pm := 1 + \frac{c_\pm}{\sqrt{n}}$ for some constants $c_+, c_- \in \mathbb{R}$. We let the weights be $(W_\ell^{\text{pre}})_{ij}(W_\ell^{\text{post}})_{ij} \overset{\text{iid}}{\sim} \mathcal{N}(0,1)$ and $c^{-1} = \mathbb{E}\,\sigma_s(g)^2$ for $g \sim \mathcal{N}(0,1)$ is the He initialization constant [55].

From here onwards, we will define the filtration $\mathcal{F}_\ell = \sigma(\{X_k\}_{k \in [\ell]})$, where $\sigma(\cdot)$ denotes the sigma-algebra generated by the random variable. Furthermore, we will define the conditional expectation $\mathbb{E}_\ell[\,\cdot\,] = \mathbb{E}[\,\cdot\,|\mathcal{F}_\ell]$.

We are interested in studying the *neural covariance*, i.e. $V_\ell^{\alpha\beta} := \frac{c}{n}\langle x_\ell^\alpha, x_\ell^\beta\rangle$ where $\ell \in [d]$ indexes the layers and $d$ is the depth. In particular, we are interested in understanding the simultaneous limit $(n, d) \to \infty$, where the ratio $t = d/n$ remains constant.

From now on, we will remove the dependence on $\ell$. Defining the residual branch as $f(X, W) := \frac{1}{\sqrt{n}}\sigma_s\left(\sqrt{\frac{c}{n}}XW^{\text{pre}}\right)W^{\text{post}}$, we can write $V^{\alpha\beta}$ as:

$$V^{\alpha\beta} = \lambda^2 X + \lambda\gamma\langle x^\alpha, f(X, W)^\beta\rangle + \lambda\gamma\langle x^\beta, f(X, W)^\alpha\rangle + \gamma^2\langle f(X, W)^\alpha, f(X, W)^\beta\rangle.$$

For the cross product terms, we get:

$$\langle x^\alpha, f(X, W)^\beta\rangle = \frac{\sqrt{c}}{n}\langle x^\alpha, \left(\sigma_s\left(XW^{\text{pre}}\right)W^{\text{post}}\right)^\beta\rangle = \frac{\sqrt{c}}{n}\sum_{i,j}^n x_i^\alpha W_{ji}^{\text{post}}\sigma_s\left(\sum_{j'} x_{j'}^\beta W_{j'j}^{\text{pre}}\right),$$

$$\langle x^\beta, f(X, W)^\alpha\rangle = \frac{\sqrt{c}}{n}\sum_{i,j}^n x_i^\beta W_{ji}^{\text{post}}\sigma_s\left(\sum_{j'} x_{j'}^\alpha W_{j'j}^{\text{pre}}\right).$$

For the term in $\gamma^2$:

$$\frac{c}{n^2}\langle(\sigma_s(XW^{\mathrm{pre}})W^{\mathrm{post}})^\alpha,(\sigma_s(XW^{\mathrm{pre}})W^{\mathrm{post}})^\beta\rangle = \frac{c}{n^2}\sum_{i=1}^n(\sigma_s(XW^{\mathrm{pre}})W^{\mathrm{post}})_{\alpha i}(\sigma_s(XW^{\mathrm{pre}})W^{\mathrm{post}})_{\beta i}$$

$$= \frac{c}{n^2}\sum_{i,j,j'=1}^n W^{\mathrm{post}}_{ji}W^{\mathrm{post}}_{j'i}\sigma_s\left(\sum_k x^\alpha_k W^{\mathrm{pre}}_{kj}\right)\sigma_s\left(\sum_{k'} x^\beta_{k'}W^{\mathrm{post}}_{k'j'}\right).$$

We will define the following terms

$$\mathcal{T}_1^{\alpha\beta} := \frac{c\sqrt{c}}{n\sqrt{n}}\sum_{i,j=1}^n W^{\mathrm{post}}_{ji}\left(x^\alpha_i\sigma_s\left(\sum_{j'}x^\beta_{j'}W^{\mathrm{pre}}_{j'j}\right) + x^\beta_i\sigma_s\left(\sum_{j'}x^\alpha_{j'}W^{\mathrm{pre}}_{j'j}\right)\right),$$

and

$$\mathcal{T}_2^{\alpha\beta} := \frac{c^2}{n^2\sqrt{n}}\sum_{i,j,j'=1}^n W^{\mathrm{post}}_{ji}W^{\mathrm{post}}_{j'i}\sigma_s\left(\sum_k x^\alpha_k W^{\mathrm{pre}}_{kj}\right)\sigma_s\left(\sum_{k'} x^\beta_{k'}W^{\mathrm{pre}}_{k'j'}\right).$$

Hence we get the following update for $V_\ell^{\alpha\beta} := \frac{c}{n}\langle x^\alpha_\ell, x^\beta_\ell\rangle$:

$$V_{\ell+1}^{\alpha\beta} = \lambda^2 V_\ell^{\alpha\beta} + \frac{\gamma\lambda}{\sqrt{n}}\mathcal{T}_1^{\alpha\beta} + \frac{\gamma^2}{\sqrt{n}}\mathcal{T}_2^{\alpha\beta}.$$

It is easy to see that the cross product terms have (conditional) mean zero. For the term with $\gamma^2$:

$$\mathbb{E}_\ell[\mathcal{T}_2^{\alpha\beta}] = \sqrt{n}\sqrt{V^{\alpha\alpha}V^{\beta\beta}}cK_1(\rho^{\alpha\beta}), \tag{23}$$

where $K_1(\rho^{\alpha\beta}) := \mathbb{E}[\sigma_s(g^\alpha)\sigma_s(g^\beta)]$ where $g^\alpha, g^\beta \sim \mathcal{N}(0,1)$ and $\mathbb{E}[g^\alpha g^\beta] = \frac{\langle x^\alpha, x^\beta\rangle}{\|x^\alpha\|\|x^\beta\|}$. Here we recall $\mathbb{E}_\ell[\cdot] = \mathbb{E}[\cdot|\mathcal{F}_\ell]$ is the conditional expectation given the sigma-algebra generated by $\mathcal{F}_\ell = \sigma(\{X_k\}_{k\in[\ell]})$.

M. Li et al. [18] showed that if the non linearity scaling exponent is $p = 1/2$, then:

$$cK_1(\rho) = \rho + \frac{\nu(\rho)}{n} + \mathcal{O}(n^{-3/2}),$$

where $\nu(\rho) = \frac{(c_+ + c_-)^2}{2\pi}\left(\sqrt{1-\rho^2} + \rho\arccos(\rho)\right)$. Using this result, and summing and subtracting the mean of $\mathcal{T}_2$:

$$V_{\ell+1}^{\alpha\beta} = \lambda^2 V_\ell^{\alpha\beta} + \gamma^2\sqrt{V^{\alpha\alpha}V^{\beta\beta}}cK_1(\rho^{\alpha\beta}) + \frac{\gamma\lambda}{\sqrt{n}}\mathcal{T}_1^{\alpha\beta} + \frac{\gamma^2}{\sqrt{n}}(\mathcal{T}_2^{\alpha\beta} - \mathbb{E}_\ell[\mathcal{T}_2^{\alpha\beta}])$$

$$= \lambda^2 V_\ell^{\alpha\beta} + \gamma^2 V_\ell^{\alpha\beta} + \gamma^2\sqrt{V^{\alpha\alpha}V^{\beta\beta}}\frac{\nu(\rho^{\alpha\beta})}{n} + \frac{\gamma\lambda}{\sqrt{n}}\mathcal{T}_1^{\alpha\beta} + \frac{\gamma^2}{\sqrt{n}}(\mathcal{T}_2^{\alpha\beta} - \mathbb{E}_\ell[\mathcal{T}_2^{\alpha\beta}]) + \mathcal{O}(n^{-3/2}).$$

Using $\lambda^2 + \gamma^2 = 1$:

$$V_{\ell+1}^{\alpha\beta} = V_\ell^{\alpha\beta} + \gamma^2\sqrt{V^{\alpha\alpha}V^{\beta\beta}}\frac{\nu(\rho^{\alpha\beta})}{n} + \frac{\gamma\lambda}{\sqrt{n}}\mathcal{T}_1^{\alpha\beta} + \frac{\gamma^2}{\sqrt{n}}(\mathcal{T}_2^{\alpha\beta} - \mathbb{E}_\ell[\mathcal{T}_2^{\alpha\beta}]) + \mathcal{O}(n^{-3/2}).$$

Furthermore, we need second order moments of $\mathcal{T}_1$ and $\mathcal{T}_2$. We derive them in the following Lemmas:

**Lemma B.1.**

$$\mathbb{E}_\ell[\mathcal{T}_1^{\alpha\beta}\mathcal{T}_1^{\delta\omega}] = V^{\alpha\delta}\sqrt{V^{\beta\beta}V^{\omega\omega}}cK_1(\rho^{\beta\omega}) + V^{\alpha\omega}\sqrt{V^{\beta\beta}V^{\delta\delta}}cK_1(\rho^{\beta\delta})$$

$$+ V^{\beta\delta}\sqrt{V^{\alpha\alpha}V^{\omega\omega}}cK_1(\rho^{\alpha\omega}) + V^{\beta\omega}\sqrt{V^{\alpha\alpha}V^{\delta\delta}}cK_1(\rho^{\alpha\delta})$$

$$= 2V^{\alpha\delta}V^{\beta\omega} + 2V^{\alpha\omega}V^{\beta\delta} + \mathcal{O}(n^{-1}).$$

*Proof.* Recall the definition of $\mathcal{T}_1^{\alpha\beta}$:

$$\mathcal{T}_1^{\alpha\beta} := \frac{c\sqrt{c}}{n\sqrt{n}} \sum_{i,j=1}^{n} W_{ji}^{\text{post}} \left( x_i^{\alpha} \sigma_s \left( \sum_{j'} x_{j'}^{\beta} W_{j'j}^{\text{pre}} \right) + x_i^{\beta} \sigma_s \left( \sum_{j'} x_{j'}^{\alpha} W_{j'j}^{\text{pre}} \right) \right),$$

We have that:

$$\mathbb{E}_{\ell}[\mathcal{T}_1^{\alpha\beta}\mathcal{T}_1^{\delta\omega}] = \frac{c^3}{n^3} \sum_{iji'j'} \mathbb{E}\left[W_{ji}W_{j'i'}\right] \mathbb{E}_{\ell} \left[ \left( x_i^{\alpha} \left\| x^{\beta} \right\| \sigma_s(g_j^{\beta}) + x_i^{\beta} \left\| x^{\alpha} \right\| \sigma_s(g_j^{\alpha}) \right) \left( x_i^{\delta} \left\| x^{\omega} \right\| \sigma_s(g_j^{\omega}) + x_i^{\omega} \left\| x^{\delta} \right\| \sigma_s(g_j^{\delta}) \right) \right]$$

$$= \frac{c^3}{n^3} \sum_{ij} \mathbb{E}_{\ell}\left[ x_i^{\alpha} x_i^{\delta} \left\| x^{\beta} \right\| \left\| x^{\omega} \right\| \sigma_s(g_j^{\beta})\sigma_s(g_j^{\omega}) + x_i^{\alpha} x_i^{\omega} \left\| x^{\beta} \right\| \left\| x^{\delta} \right\| \sigma_s(g_j^{\beta})\sigma_s(g_j^{\delta}) \right.$$

$$\left. + x_i^{\beta} x_i^{\delta} \left\| x^{\alpha} \right\| \left\| x^{\omega} \right\| \sigma_s(g_j^{\alpha})\sigma_s(g_j^{\omega}) + x_i^{\beta} x_i^{\omega} \left\| x^{\alpha} \right\| \left\| x^{\delta} \right\| \sigma_s(g_j^{\alpha})\sigma_s(g_j^{\delta}) \right]$$

$$= V^{\alpha\delta}\sqrt{V^{\beta\beta}V^{\omega\omega}}cK_1(\rho^{\beta\omega}) + V^{\alpha\omega}\sqrt{V^{\beta\beta}V^{\delta\delta}}cK_1(\rho^{\beta\delta})$$

$$+ V^{\beta\delta}\sqrt{V^{\alpha\alpha}V^{\omega\omega}}cK_1(\rho^{\alpha\omega}) + V^{\beta\omega}\sqrt{V^{\alpha\alpha}V^{\delta\delta}}cK_1(\rho^{\alpha\delta})$$

$\square$

**Lemma B.2.**

$$\mathbb{E}_{\ell}[\mathcal{T}_2^{\alpha\beta}] = c\sqrt{n}\sqrt{V^{\alpha\alpha}V^{\beta\beta}}K_1(\rho^{\alpha\beta}) = \sqrt{n}V^{\alpha\beta} + \frac{1}{\sqrt{n}}\sqrt{V^{\alpha\alpha}V^{\beta\beta}}\nu(\rho) + \mathcal{O}(n^{-1}),$$

$$\mathbb{E}_{\ell}[\mathcal{T}_2^{\alpha\beta}\mathcal{T}_2^{\delta\omega}] - \mathbb{E}_{\ell}[\mathcal{T}_2^{\alpha\beta}]\mathbb{E}[\mathcal{T}_2^{\delta\omega}] = 2V^{\alpha\delta}V^{\beta\omega} + 2V^{\alpha\omega}V^{\beta\delta} + \mathcal{O}(n^{-1}).$$

*Proof.* Recall the definition of $\mathcal{T}_2^{\alpha\beta}$:

$$\mathcal{T}_2^{\alpha\beta} := \frac{c^2}{n^2\sqrt{n}} \sum_{i,j,j'=1}^{n} W_{ji}^{\text{post}}W_{j'i}^{\text{post}}\sigma_s \left( \sum_k x_k^{\alpha}W_{kj}^{\text{pre}} \right) \sigma_s \left( \sum_{k'} x_{k'}^{\beta}W_{k'j'}^{\text{pre}} \right)$$

$$= \frac{c^2}{n^2\sqrt{n}} \left\| x^{\alpha} \right\| \left\| x^{\beta} \right\| \sum_{i,j,j'=1}^{n} W_{ji}^{\text{post}}W_{j'i}^{\text{post}}\sigma_s(g_j^{\alpha})\sigma_s(g_{j'}^{\beta})$$

For the mean, using the independence between $W^{\text{pre}}$ and $W^{\text{post}}$, we have that:

$$\mathbb{E}_{\ell}\left[\mathcal{T}_2^{\alpha\beta}\right] = \frac{c^2}{n^2\sqrt{n}} \left\| x^{\alpha} \right\| \left\| x^{\beta} \right\| \sum_{i,j,j'=1}^{n} \mathbb{E}\left[W_{ji}^{\text{post}}W_{j'i}^{\text{post}}\right] \mathbb{E}\left[\sigma_s(g_j^{\alpha})\sigma_s(g_{j'}^{\beta})\right]$$

$$= \frac{c^2}{n\sqrt{n}} \left\| x^{\alpha} \right\| \left\| x^{\beta} \right\| \sum_{j=1}^{n} \mathbb{E}\left[\sigma_s(g_j^{\alpha})\sigma_s(g_j^{\beta})\right]$$

$$= \frac{c^2}{\sqrt{n}} \left\| x^{\alpha} \right\| \left\| x^{\beta} \right\| K_1(\rho^{\alpha\beta})$$

$$= c\sqrt{n}\sqrt{V^{\alpha\alpha}V^{\beta\beta}}K_1(\rho^{\alpha\beta}),$$

which is the desired result. The final expression is the result of the aforementioned expansion for $K_1$ [18]. For the covariance, we have that:

$$\mathbb{E}_{\ell}[\mathcal{T}_2^{\alpha\beta}\mathcal{T}_2^{\delta\omega}] = \frac{c^4}{n^5} \left\| x^{\alpha} \right\| \left\| x^{\beta} \right\| \left\| x^{\delta} \right\| \left\| x^{\omega} \right\| \sum_{i,j,j',i',k,k'} \mathbb{E}\left[W_{ji}^{\text{post}}W_{j'i}^{\text{post}}W_{ki'}^{\text{post}}W_{k'i'}^{\text{post}}\right] \mathbb{E}\left[\sigma_s(g_j^{\alpha})\sigma_s(g_{j'}^{\beta})\sigma_s(g_k^{\delta})\sigma_s(g_{k'}^{\omega})\right]$$

$$= \frac{c^2}{n^3}\sqrt{V^{\alpha\alpha}V^{\beta\beta}V^{\delta\delta}V^{\omega\omega}} \sum_{i,j,j',i',k,k'} (\delta_{jj'}\delta_{kk'} + \delta_{jk}\delta_{ii'}\delta_{j'k'} + \delta_{jk'}\delta_{ii'}\delta_{j'k}) \mathbb{E}\left[\sigma_s(g_j^{\alpha})\sigma_s(g_{j'}^{\beta})\sigma_s(g_k^{\delta})\sigma_s(g_{k'}^{\omega})\right].$$

Let's look at each term of the sum separately. For the first term we have that:

$$\sum_{i,j,j',i',k,k'} \delta_{jj'}\delta_{kk'} \mathbb{E}\left[\sigma_s(g_j^\alpha)\sigma_s(g_{j'}^\beta)\sigma_s(g_k^\delta)\sigma_s(g_{k'}^\omega)\right]$$

$$= n^2 \sum_{j,k} \mathbb{E}\left[\sigma_s(g_j^\alpha)\sigma_s(g_j^\beta)\sigma_s(g_k^\delta)\sigma_s(g_k^\omega)\right]$$

$$= n^2 \sum_j \mathbb{E}\left[\sigma_s(g_j^\alpha)\sigma_s(g_j^\beta)\sigma_s(g_j^\delta)\sigma_s(g_j^\omega)\right] + n^2 \sum_{j\neq k} \mathbb{E}\left[\sigma_s(g_j^\alpha)\sigma_s(g_j^\beta)\right]\mathbb{E}\left[\sigma_s(g_k^\delta)\sigma_s(g_k^\omega)\right]$$

$$= n^3 K_2(\rho^{\alpha\beta\delta\omega}) + n^3(n-1)K_1(\rho^{\alpha\beta})K_1(\rho^{\delta\omega}),$$

where we have defined the fourth moment of the shaped activation: $K_2(\rho^{\alpha\beta\delta\omega}) := \mathbb{E}\left[\sigma_s(g^\alpha)\sigma_s(g^\beta)\sigma_s(g^\delta)\sigma_s(g^\omega)\right]$ for which it holds that [18, Lemma C.2]:

$$K_2(\rho^{\alpha\beta\delta\omega}) = \mathbb{E}[g^\alpha g^\beta g^\delta g^\omega] + \mathcal{O}(n^{-1/2}) = \rho^{\alpha\beta}\rho^{\delta\omega} + \rho^{\alpha\delta}\rho^{\beta\omega} + \rho^{\alpha\omega}\rho^{\beta\delta} + \mathcal{O}(n^{-1/2}).$$

The other two summands can be solved similarly:

$$\sum_{i,j,j',i',k,k'} \delta_{jk}\delta_{ii'}\delta_{j'k'} \mathbb{E}\left[\sigma_s(g_j^\alpha)\sigma_s(g_{j'}^\beta)\sigma_s(g_k^\delta)\sigma_s(g_{k'}^\omega)\right] = n^2 K_2(\rho^{\alpha\beta\delta\omega}) + n^2(n-1)K_1(\rho^{\alpha\delta})K_1(\rho^{\beta\omega}),$$

and

$$\sum_{i,j,j',i',k,k'} \delta_{jk'}\delta_{ii'}\delta_{j'k} \mathbb{E}\left[\sigma_s(g_j^\alpha)\sigma_s(g_{j'}^\beta)\sigma_s(g_k^\delta)\sigma_s(g_{k'}^\omega)\right] = n^2 K_2(\rho^{\alpha\beta\delta\omega}) + n^2(n-1)K_1(\rho^{\alpha\omega})K_1(\rho^{\beta\delta}).$$

Hence, summing the three terms:

$$\mathbb{E}_\ell[\mathcal{T}_2^{\alpha\beta}\mathcal{T}_2^{\delta\omega}] = c^2\sqrt{V^{\alpha\alpha}V^{\beta\beta}V^{\delta\delta}V^{\omega\omega}}\Big(K_2(\rho^{\alpha\beta\delta\omega}) + (n-1)K_1(\rho^{\alpha\beta})K_1(\rho^{\delta\omega})$$

$$+ \frac{1}{n}\left(K_2(\rho^{\alpha\beta\delta\omega}) + (n-1)K_1(\rho^{\alpha\delta})K_1(\rho^{\beta\omega})\right) + \frac{1}{n}\left(K_2(\rho^{\alpha\beta\delta\omega}) + (n-1)K_1(\rho^{\alpha\omega})K_1(\rho^{\beta\delta})\right)\Big)$$

$$= c^2\sqrt{V^{\alpha\alpha}V^{\beta\beta}V^{\delta\delta}V^{\omega\omega}}\Big(K_2(\rho^{\alpha\beta\delta\omega}) + nK_1(\rho^{\alpha\beta})K_1(\rho^{\delta\omega}) - K_1(\rho^{\alpha\beta})K_1(\rho^{\delta\omega})$$

$$+ K_1(\rho^{\alpha\delta})K_1(\rho^{\beta\omega}) + K_1(\rho^{\alpha\omega})K_1(\rho^{\beta\delta})\Big) + \mathcal{O}(n^{-1}).$$

Now, subtracting $\mathbb{E}_\ell[\mathcal{T}_2^{\alpha\beta}]\mathbb{E}_\ell[\mathcal{T}_2^{\delta\omega}]$, using the aforementioned expansions for $K_1$ and $K_2$:

$$\mathbb{E}_\ell[\mathcal{T}_2^{\alpha\beta}\mathcal{T}_2^{\delta\omega}] - \mathbb{E}_\ell[\mathcal{T}_2^{\alpha\beta}]\mathbb{E}_\ell[\mathcal{T}_2^{\delta\omega}] = c^2\sqrt{V^{\alpha\alpha}V^{\beta\beta}V^{\delta\delta}V^{\omega\omega}}\left(\rho^{\alpha\beta}\rho^{\delta\omega} + \rho^{\alpha\delta}\rho^{\beta\omega} + \rho^{\alpha\omega}\rho^{\beta\delta}\right)$$

$$- V^{\alpha\beta}V^{\delta\omega} + V^{\alpha\delta}V^{\beta\omega} + V^{\alpha\omega}V^{\beta\delta} + \mathcal{O}(n^{-1})$$

$$= 2V^{\alpha\delta}V^{\beta\omega} + 2V^{\alpha\omega}V^{\beta\delta} + \mathcal{O}(n^{-1}),$$

where in the final step we have used the fact that $c = 1 + \mathcal{O}(n^{-1/2})$. This completes the proof. □

Finally we also have that $\mathcal{T}_1$ and $\mathcal{T}_2$ are uncorrelated, i.e.

**Lemma B.3.**

$$\mathbb{E}_\ell[\mathcal{T}_1^{\alpha\beta}\mathcal{T}_2^{\delta\omega}] = 0.$$

*Proof.* It is easy to see that $\mathbb{E}_\ell\mathcal{T}_1^{\alpha\beta}\mathcal{T}_2^{\delta\omega}$ involve odd standard Gaussian moments (in particular third order moments), which vanish due to the parity of the centered Gaussian measure. □

**Theorem 3.2.** *Let $X_\ell$ be the hidden layers of a ResNet defined in Eq. 4 with $\lambda^2 + \gamma^2 = 1$, where both $\lambda$ and $\gamma$ do not depend on $d, n$. Then the feature covariance $V_\ell$ converges to the solution of the following SDE (in the sense of Definition 3.1)*

$$dV_t = b_{res}(V_t)\,dt + \Sigma_{res}(V_t)^{1/2}\,dB_t, \quad V_0 = \frac{1}{n}X_0 X_0^\top, \tag{5}$$

*where $b_{res}(V) = \gamma^2 b_{ReLU}(V) = \gamma^2[\nu(\rho^{\alpha\beta})\sqrt{V^{\alpha\alpha}V^{\beta\beta}}]_{\alpha\leq\beta}$ with $\rho^{\alpha\beta} = V^{\alpha\beta}(V^{\alpha\alpha}V^{\beta\beta})^{-1/2}$ and*

$$\nu(\rho) = \frac{(c_+ - c_-)^2}{2\pi}\left(\sqrt{1-\rho^2} - \rho\arccos\rho\right), \tag{6}$$

*furthermore, $\Sigma_{res}(V) = 2\gamma^2\Sigma_{lin}(V) = 2\gamma^2[V^{\alpha\delta}V^{\beta\omega} + V^{\alpha\omega}V^{\beta\delta}]_{\alpha\leq\beta,\delta\leq\omega}$.*

*Proof.* We know that:

$$V_{\ell+1}^{\alpha\beta} = \lambda^2 V_\ell^{\alpha\beta} + \frac{\gamma\lambda}{\sqrt{n}}\mathcal{T}_1^{\alpha\beta} + \frac{\gamma^2}{\sqrt{n}}\mathcal{T}_2^{\alpha\beta}.$$

Using $\lambda^2 + \gamma^2 = 1$, and summing and subtracting the mean of $\mathcal{T}_2$, we have that:

$$V_{\ell+1}^{\alpha\beta} = V_\ell^{\alpha\beta} - \gamma^2 V_\ell^{\alpha\beta} + \frac{\gamma^2}{\sqrt{n}}\mathbb{E}_\ell\left[\mathcal{T}_2^{\alpha\beta}\right] + \frac{\gamma\lambda}{\sqrt{n}}\mathcal{T}_1^{\alpha\beta} + \frac{\gamma^2}{\sqrt{n}}\left(\mathcal{T}_2^{\alpha\beta} - \mathbb{E}_\ell\left[\mathcal{T}_2^{\alpha\beta}\right]\right).$$

Using Lemma B.2 for the mean of $\mathcal{T}_2$, we have that:

$$V_{\ell+1}^{\alpha\beta} = V_\ell^{\alpha\beta} + \frac{1}{n}\sqrt{V^{\alpha\alpha}V^{\beta\beta}}\nu(\rho) + \frac{\gamma\lambda}{\sqrt{n}}\mathcal{T}_1^{\alpha\beta} + \frac{\gamma^2}{\sqrt{n}}\left(\mathcal{T}_2^{\alpha\beta} - \mathbb{E}_\ell\left[\mathcal{T}_2^{\alpha\beta}\right]\right) + \mathcal{O}(n^{-3/2}),$$

which gives us an expression in the right Markov Chain form with drift term $\sqrt{V^{\alpha\alpha}V^{\beta\beta}}\nu(\rho)$. For the diffusion term, we need to compute the covariance between two different neural covariances $V_{\ell+1}^{\alpha\beta}, V_{\ell+1}^{\delta\omega}$. Using Lemmas B.1 to B.3, we have

$$\begin{aligned}
\text{Cov}_\ell\left(V_{\ell+1}^{\alpha\beta}, V_{\ell+1}^{\delta\omega}\right) &= \mathbb{E}_\ell\left[\left(\lambda\gamma\mathcal{T}_1^{\alpha\beta} + \gamma^2\mathcal{T}_2^{\alpha\beta} - \gamma^2\mathbb{E}_\ell[\mathcal{T}_2^{\alpha\beta}]\right)\left(\lambda\gamma\mathcal{T}_1^{\delta\omega} + \gamma^2\mathcal{T}_2^{\delta\omega} - \gamma^2\mathbb{E}_\ell[\mathcal{T}_2^{\delta\omega}]\right)\right] \\
&= \lambda^2\gamma^2\mathbb{E}_\ell\left[\mathcal{T}_1^{\alpha\beta}\mathcal{T}_1^{\delta\omega}\right] + \gamma^4\mathbb{E}_\ell\left[\mathcal{T}_2^{\alpha\beta}\mathcal{T}_2^{\delta\omega}\right] - \gamma^4\mathbb{E}_\ell\left[\mathcal{T}_2^{\alpha\beta}\right]\mathbb{E}_\ell\left[\mathcal{T}_2^{\delta\omega}\right] \\
&= 2\lambda^2\gamma^2(V^{\alpha\delta}V^{\beta\omega} + V^{\alpha\omega}V^{\beta\delta}) + 2\gamma^4(V^{\alpha\delta}V^{\beta\omega} + V^{\alpha\omega}V^{\beta\delta}) + \mathcal{O}(n^{-1}) \\
&= 2\gamma^2(V^{\alpha\delta}V^{\beta\omega} + V^{\alpha\omega}V^{\beta\delta}) + \mathcal{O}(n^{-1}),
\end{aligned}$$

where we use $\text{Cov}_\ell$ to denote the covariance conditioned on the sigma-algebra $\mathcal{F}_\ell = \sigma(\{X_k\}_{k\in[\ell]})$.

Finally, applying Proposition A.2, we get the desired result. $\square$

## C  SDE for Softmax-based Attention

### C.1  Dot-Product Attention

Dot-product attention applies the Softmax row-wise to the following matrix:

$$Y_\ell = \frac{1}{n}X_\ell\underbrace{W_\ell^K W_\ell^{Q,\top}}_{W_\ell^B}X_\ell^\top,$$

where $W_\ell^K, W_\ell^Q \in \mathbb{R}^{n\times n_k}$ are Gaussian matrices with unit variance entries, and $n_k$ is the queries and keys' dimension. Here we study the first two moments of $Y_\ell$. In particular, note that $\mathbb{E}_\ell[Y_\ell] = 0$, where $\mathbb{E}_\ell[\cdot] = \mathbb{E}[\cdot|\mathcal{F}_\ell]$ denotes the conditional expectation given the sigma-algebra generated by $\mathcal{F}_\ell = \sigma(\{X_k\}_{k\in[\ell]})$.

For the second moment we have the following Lemma.

**Lemma C.1.** *Let*

$$Y_\ell = \frac{1}{n}X_\ell W_\ell^K W_\ell^{Q,\top} X_\ell^\top,$$

*be the dot-product attention parametrized by $W_\ell^K, W_\ell^Q \in \mathbb{R}^{n\times n_k}$. Then:*

$$\mathbb{E}_\ell[Y^{\alpha\beta}Y^{\delta\omega}] = n_k V^{\alpha\delta}V^{\beta\omega}. \tag{24}$$

*Proof.*

$$\mathbb{E}_\ell[Y^{\alpha\beta}Y^{\delta\omega}] = \frac{1}{n^2}\sum_{kk'jj'}X_k^\alpha X_{k'}^\beta X_j^\delta X_{j'}^\omega\mathbb{E}[W_{kk'}^B W_{jj'}^B].$$

For the expectation, we have that:

$$\mathbb{E}[W^B_{kk'}W^B_{jj'}] = \sum_{ii'}^{n_k} \mathbb{E}\left[W^K_{ki}W^K_{ji'}W^Q_{k'i}W^Q_{j'i'}\right]$$

$$= \sum_{ii'} \mathbb{E}\left[W^K_{ki}W^K_{ji'}\right]\mathbb{E}\left[W^Q_{k'i}W^Q_{j'i'}\right]$$

$$= \sum_{ii'} \delta_{kj}\delta_{k'j'}\delta_{ii'}$$

$$= n_k\delta_{kj}\delta_{k'j'}\,,$$

where we recall $W^B = W^K W^{Q,\top}$.

Hence:

$$\mathbb{E}_\ell[Y^{\alpha\beta}Y^{\delta\omega}] = \frac{n_k}{n^2}\sum_{kk'} X^\alpha_k X^\beta_{k'} X^\delta_k X^\omega_{k'} = n_k V^{\alpha\delta}V^{\beta\omega}\,.$$

$\square$

## C.2  Shaping the Softmax

Recall that we have the following model:

$$X_{\ell+1} = \lambda X_\ell + \gamma A_\ell X_\ell \frac{1}{\sqrt{n}}W^V_\ell$$

For the above model, $V_\ell$ has the following form:

$$V_{\ell+1} = \lambda^2 V_\ell + \frac{\lambda\gamma}{n\sqrt{n}}\left(X_\ell W^\top_\ell X^\top_\ell A^\top_\ell + A_\ell X_\ell W_\ell X^\top_\ell\right) + \frac{\gamma^2}{n^2}A_\ell X_\ell W_\ell W^\top_\ell X^\top_\ell A^\top_\ell\,.$$

In order to have infinitesimal updates in $V_\ell$, we would intuitively like the attention matrix to be of the form $A^{\alpha\beta} = \delta_{\alpha\beta} + \mathcal{O}(n^{-1})$. In the case of the usual Softmax based attention, we have:

$$A_\ell := \text{Softmax}(\tau^{-1}Y_\ell),$$

where $\tau^{-1}$ is a temperature parameter that regulates the entropy of the resulting categorical distribution; $\tau$ is often chosen to be large (scale as a power of $n$ or $d$), as low temperature results in unstable training.

To transform the Softmax into the desired form of $A^{\alpha\beta} = \delta_{\alpha\beta} + \mathcal{O}(n^{-1})$, we first center the attention matrix around the identity matrix:

$$A_\ell = I + \text{Softmax}(\tau^{-1}Y_\ell),$$

and then examine the Taylor-expansion of the Softmax w.r.t $\tau^{-1}$:

$$A_\ell = I + \frac{1}{m}11^\top + \mathcal{O}(\tau^{-1})\,. \tag{25}$$

The first few terms of the expansion provides a good approximation of $A_\ell$ when $\tau$ is large. Observe that for fixed $m$, the zero order term $\frac{1}{m}11^\top$ is not of $\mathcal{O}(n^{-1})$. In particular, suppose that we use the attention model in Eq. 25. We can show that the expectation of the term in $\gamma^2$ (responsible for the drift) has the following form:

$$\gamma^2\sum_{\delta,\omega=1}^m \mathbb{E}_\ell[A^{\alpha\delta}A^{\beta\omega}]V^{\delta\omega} = \gamma^2 V^{\alpha\beta} + \frac{\gamma^2}{m}\sum_\omega V^{\alpha\omega} + \frac{\gamma^2}{m}\sum_\delta V^{\delta\beta} + \frac{\gamma^2}{m^2}\sum_{\delta\omega} V^{\delta\omega} + \mathcal{O}(\tau^{-1}),$$

where the terms scaled by $\frac{1}{m}$ leads to large incremental updates in expected value of $V_\ell$ w.r.t the previous layer, and precludes the possibility of convergence to an SDE. Hence, we choose to re-center the Softmax by removing the term $\frac{1}{m}11^\top$, as follows:

$$A_\ell = I + \text{Softmax}(\tau^{-1}Y_\ell) - \frac{1}{m}11^\top,$$

which admits the following Taylor-expansion:

$$[\text{Softmax}(\tau^{-1}Y_\ell)-m^{-1}11^\top] = \frac{1}{\tau m}[Y^{\alpha\beta}-\overline{Y^\alpha}]_{\alpha,\beta}+\frac{1}{2\tau^2 m}\left[(Y^{\alpha\beta}-\overline{Y^\alpha})^2-(\overline{(Y^\alpha)^2}-\overline{Y^\alpha}^2)\right]_{\alpha,\beta}+O(\tau^{-3}),$$

where $\overline{Y^\alpha} := \frac{1}{m}\sum_\nu Y^{\alpha\nu}$, and $\overline{(Y^\alpha)^2} := \frac{1}{m}\sum_\nu (Y^{\alpha\nu})^2$.

Hence, up to third order:

$$A_\ell = I + \frac{1}{\tau m}[Y^{\alpha\beta} - \overline{Y^\alpha}]_{\alpha,\beta} + \frac{1}{2\tau^2 m}\left[(Y^{\alpha\beta} - \overline{Y^\alpha})^2 - (\overline{(Y^\alpha)^2} - \overline{Y^\alpha}^2)\right]_{\alpha\beta} + O(\tau^{-3}).$$

For sufficiently large $\tau$, the formulation above allows infinitesimal updates in $V_\ell$, and as we will show rigorously in the rest of this section, permits convergence to an SDE.

## C.3 Lemmas on Moments of Shaped Attention

Define:

$$F_1^{\alpha\beta} = Y^{\alpha\beta} - \overline{Y^\alpha},$$
$$F_2^{\alpha\beta} = (Y^{\alpha\beta} - \overline{Y^\alpha})^2 - (\overline{(Y^\alpha)^2} - \overline{Y^\alpha}^2).$$

Hence, $A_\ell$ can be written as:

$$A_\ell^{\alpha\beta} = \delta_{\alpha\beta} + \frac{1}{\tau m}F_1^{\alpha\beta} + \frac{1}{2\tau^2 m}F_2^{\alpha\beta} + O(\tau^{-3}).$$

We now compute the moments of $A_\ell$. We define the following quantities:

$$S_1^{\alpha\delta,\beta\omega} := \frac{1}{n_k}\mathbb{E}_\ell(Y^{\alpha\delta} - \overline{Y^\alpha})(Y^{\beta\omega} - \overline{Y^\beta}) = \frac{1}{n_k}\mathbb{E}_\ell F_1^{\alpha\delta}F_1^{\beta\omega},$$

$$S_2^{\alpha\delta} := \frac{1}{n_k}\mathbb{E}_\ell\left[(Y^{\alpha\delta} - \overline{Y^\alpha})^2 - (\overline{(Y^\alpha)^2} - \overline{Y^\alpha}^2)\right] = \frac{1}{n_k}\mathbb{E}_\ell F_2^{\alpha\delta},$$

where we recall $\mathbb{E}_\ell[\,\cdot\,] = \mathbb{E}[\,\cdot\,|\mathcal{F}_\ell]$ is the conditional expectation given the sigma-algebra generated by $\mathcal{F}_\ell = \sigma(\{X_k\}_{k\in[\ell]})$.

**Lemma C.2** (Moments of Taylor Expansion).

$$S_1^{\alpha\delta,\beta\omega} = V^{\alpha\beta}\left(V^{\delta\omega} - V^{\delta\bar{x}} - V^{\omega\bar{x}} + V^{\bar{x}\bar{x}}\right),$$
$$S_2^{\alpha\delta} = V^{\alpha\alpha}\left(V^{\delta\delta} - 2V^{\delta\bar{x}} + 2V^{\bar{x}\bar{x}} - \bar{V}\right),$$

*where $\bar{V} = \frac{1}{m}\sum_\nu V^{\nu\nu}$ and $\bar{x} = \frac{1}{m}\sum_\nu x^\nu$ is the average token.*

*Proof.* Using Lemma C.1 and linearity of expectation:

$$S_1^{\alpha\delta,\beta\omega} = \frac{1}{n_k}\left(\mathbb{E}_\ell[Y^{\alpha\delta}Y^{\beta\omega}] - \mathbb{E}_\ell[Y^{\alpha\delta}\overline{Y^\beta}] - \mathbb{E}_\ell[Y^{\beta\omega}\overline{Y^\alpha}] + \mathbb{E}_\ell[\overline{Y^\alpha}\overline{Y^\beta}]\right)$$
$$= \left(V^{\alpha\beta}V^{\delta\omega} - V^{\alpha\beta}V^{\delta\bar{x}} - V^{\alpha\beta}V^{\omega\bar{x}} + V^{\alpha\beta}V^{\bar{x}\bar{x}}\right)$$
$$= V^{\alpha\beta}\left(V^{\delta\omega} - V^{\delta\bar{x}} - V^{\omega\bar{x}} + V^{\bar{x}\bar{x}}\right),$$

and:

$$S_2^{\alpha\delta} = \frac{1}{n_k}\left(\mathbb{E}_\ell\left[(Y^{\alpha\delta} - \overline{Y^\alpha})^2\right] - \mathbb{E}_\ell\left[(\overline{(Y^\alpha)^2} - \overline{Y^\alpha}^2)\right]\right)$$
$$= \frac{1}{n_k}\left(\mathbb{E}_\ell[(Y^{\alpha\delta})^2] - 2\mathbb{E}_\ell[Y^{\alpha\delta}\overline{Y^\alpha}] + 2\mathbb{E}_\ell[\overline{Y^\alpha}^2] - \mathbb{E}_\ell[\overline{(Y^\alpha)^2}]\right)$$
$$= \left(V^{\alpha\alpha}V^{\delta\delta} - 2V^{\alpha\alpha}V^{\delta\bar{x}} + 2V^{\alpha\alpha}V^{\bar{x}\bar{x}} - V^{\alpha\alpha}\bar{V}\right)$$
$$= V^{\alpha\alpha}\left(V^{\delta\delta} - 2V^{\delta\bar{x}} + 2V^{\bar{x}\bar{x}} - \bar{V}\right),$$

where $\bar{V} = \frac{1}{m}\sum_\beta V^{\beta\beta}$. $\qquad\square$

**Lemma C.3.**

$$\mathbb{E}_\ell[A^{\alpha\delta}A^{\beta\omega}] = \delta_{\alpha\delta}\delta_{\beta\omega} + \frac{n_k}{\tau^2 m^2}S_1^{\alpha\delta,\beta\omega} + \frac{n_k}{2\tau^2 m}(\delta_{\beta\omega}S_2^{\alpha\delta} + \delta_{\alpha\delta}S_2^{\beta\omega}) + O(n_k\tau^{-3})\,.$$

*Proof.*

$$
\begin{aligned}
\mathbb{E}_\ell[A^{\alpha\delta}A^{\beta\omega}] &= \delta_{\alpha\delta}\delta_{\beta\omega} + \frac{\delta_{\beta\omega}}{\tau m}\mathbb{E}_\ell[Y^{\alpha\delta} - \overline{Y^\alpha}] + \frac{\delta_{\alpha\delta}}{\tau m}\mathbb{E}_\ell[Y^{\beta\omega} - \overline{Y^\beta}] + \frac{1}{\tau^2 m^2}\mathbb{E}_\ell(Y^{\alpha\delta} - \overline{Y^\alpha})(Y^{\beta\omega} - \overline{Y^\beta}) \\
&\quad + \frac{\delta_{\beta\omega}}{2\tau^2 m}\mathbb{E}_\ell\left[(Y^{\alpha\delta} - \overline{Y^\alpha})^2 - (\overline{(Y^\alpha)^2} - \overline{Y^\alpha}^2)\right] \\
&\quad + \frac{\delta_{\alpha\delta}}{2\tau^2 m}\mathbb{E}_\ell\left[(Y^{\beta\omega} - \overline{Y^\beta})^2 - (\overline{(Y^\beta)^2} - \overline{Y^\beta}^2)\right] + O(n_k\tau^{-3}) \\
&= \delta_{\alpha\delta}\delta_{\beta\omega} + \frac{n_k}{\tau^2 m^2}S_1^{\alpha\delta,\beta\omega} + \frac{n_k}{2\tau^2 m}(\delta_{\beta\omega}S_2^{\alpha\delta} + \delta_{\alpha\delta}S_2^{\beta\omega}) + O(n_k\tau^{-3})\,.
\end{aligned}
$$

$\square$

**Lemma C.4.**

$$
\begin{aligned}
\mathbb{E}_\ell\left[A^{\alpha\alpha'}A^{\beta\beta'}A^{\delta\delta'}A^{\omega\omega'}\right] &= \delta_{\alpha\alpha'}\delta_{\beta\beta'}\delta_{\delta\delta'}\delta_{\omega\omega'} \\
&\quad + \frac{n_k}{\tau^2 m^2}\Big(\delta_{\alpha\alpha'}\delta_{\beta\beta'}S_1^{\delta\delta',\omega\omega'} + \delta_{\alpha\alpha'}\delta_{\delta\delta'}S_1^{\beta\beta',\omega\omega'} + \delta_{\alpha\alpha'}\delta_{\omega\omega'}S_1^{\beta\beta',\delta\delta'} \\
&\qquad + \delta_{\beta\beta'}\delta_{\delta\delta'}S_1^{\alpha\alpha',\omega\omega'} + \delta_{\beta\beta'}\delta_{\omega\omega'}S_1^{\alpha\alpha',\delta\delta'} + \delta_{\omega\omega'}\delta_{\delta\delta'}S_1^{\alpha\alpha',\beta\beta'}\Big) \\
&\quad + \frac{n_k}{2\tau^2 m}\Big(\delta_{\alpha\alpha'}\delta_{\beta\beta'}\delta_{\delta\delta'}S_2^{\omega\omega'} + \delta_{\alpha\alpha'}\delta_{\beta\beta'}\delta_{\omega\omega'}S_2^{\delta\delta'} \\
&\qquad + \delta_{\alpha\alpha'}\delta_{\omega\omega'}\delta_{\delta\delta'}S_2^{\beta\beta'} + \delta_{\omega\omega'}\delta_{\beta\beta'}\delta_{\delta\delta'}S_2^{\alpha\alpha'}\Big) + O(n_k\tau^{-3})\,.
\end{aligned}
$$

*Proof.*

$$
\begin{aligned}
\mathbb{E}_\ell\left[A^{\alpha\alpha'}A^{\beta\beta'}A^{\delta\delta'}A^{\omega\omega'}\right] &= \mathbb{E}_\ell\Bigg[\left(\delta_{\alpha\alpha'} + \frac{1}{\tau m}F_1^{\alpha\alpha'} + \frac{1}{2\tau^2 m}F_2^{\alpha\alpha'}\right)\left(\delta_{\beta\beta'} + \frac{1}{\tau m}F_1^{\beta\beta'} + \frac{1}{2\tau^2 m}F_2^{\beta\beta'}\right) \\
&\qquad\quad \left(\delta_{\delta\delta'} + \frac{1}{\tau m}F_1^{\delta\delta'} + \frac{1}{2\tau^2 m}F_2^{\delta\delta'}\right)\left(\delta_{\omega\omega'} + \frac{1}{\tau m}F_1^{\omega\omega'} + \frac{1}{2\tau^2 m}F_2^{\omega\omega'}\right)\Bigg] + O(\tau^{-3}) \\
&= \mathbb{E}_\ell\Bigg[\delta_{\alpha\alpha'}\delta_{\beta\beta'}\delta_{\delta\delta'}\delta_{\omega\omega'} + \frac{1}{\tau^2 m^2}\delta_{\alpha\alpha'}\delta_{\beta\beta'}F_1^{\delta\delta'}F_1^{\omega\omega'} + \frac{1}{\tau^2 m^2}\delta_{\alpha\alpha'}\delta_{\delta\delta'}F_1^{\beta\beta'}F_1^{\omega\omega'} \\
&\quad + \frac{1}{\tau^2 m^2}\delta_{\alpha\alpha'}\delta_{\omega\omega'}F_1^{\delta\delta'}F_1^{\beta\beta'} + \frac{1}{\tau^2 m^2}\delta_{\beta\beta'}\delta_{\delta\delta'}F_1^{\alpha\alpha'}F_1^{\omega\omega'} \\
&\quad + \frac{1}{\tau^2 m^2}\delta_{\beta\beta'}\delta_{\omega\omega'}F_1^{\alpha\alpha'}F_1^{\delta\delta'} + \frac{1}{\tau^2 m^2}\delta_{\delta\delta'}\delta_{\omega\omega'}F_1^{\alpha\alpha'}F_1^{\beta\beta'} \\
&\quad + \frac{1}{2\tau^2 m}\delta_{\alpha\alpha'}\delta_{\beta\beta'}\delta_{\delta\delta'}F_2^{\omega\omega'} + \frac{1}{2\tau^2 m}\delta_{\alpha\alpha'}\delta_{\beta\beta'}\delta_{\omega\omega'}F_2^{\delta\delta'} \\
&\quad + \frac{1}{2\tau^2 m}\delta_{\alpha\alpha'}\delta_{\omega\omega'}\delta_{\delta\delta'}F_2^{\beta\beta'} + \frac{1}{2\tau^2 m}\delta_{\omega\omega'}\delta_{\beta\beta'}\delta_{\delta\delta'}F_2^{\alpha\alpha'}\Bigg] + O(\tau^{-3})\,.
\end{aligned}
$$

Using the linearity of expectation:

$$
\begin{aligned}
\mathbb{E}_\ell\left[A^{\alpha\alpha'}A^{\beta\beta'}A^{\delta\delta'}A^{\omega\omega'}\right] &= \delta_{\alpha\alpha'}\delta_{\beta\beta'}\delta_{\delta\delta'}\delta_{\omega\omega'} \\
&\quad + \frac{n_k}{\tau^2 m^2}\Big(\delta_{\alpha\alpha'}\delta_{\beta\beta'}S_1^{\delta\delta',\omega\omega'} + \delta_{\alpha\alpha'}\delta_{\delta\delta'}S_1^{\beta\beta',\omega\omega'} + \delta_{\alpha\alpha'}\delta_{\omega\omega'}S_1^{\beta\beta',\delta\delta'} \\
&\qquad + \delta_{\beta\beta'}\delta_{\delta\delta'}S_1^{\alpha\alpha',\omega\omega'} + \delta_{\beta\beta'}\delta_{\omega\omega'}S_1^{\alpha\alpha',\delta\delta'} + \delta_{\omega\omega'}\delta_{\delta\delta'}S_1^{\alpha\alpha',\beta\beta'}\Big) \\
&\quad + \frac{n_k}{2\tau^2 m}\Big(\delta_{\alpha\alpha'}\delta_{\beta\beta'}\delta_{\delta\delta'}S_2^{\omega\omega'} + \delta_{\alpha\alpha'}\delta_{\beta\beta'}\delta_{\omega\omega'}S_2^{\delta\delta'} \\
&\qquad + \delta_{\alpha\alpha'}\delta_{\omega\omega'}\delta_{\delta\delta'}S_2^{\beta\beta'} + \delta_{\omega\omega'}\delta_{\beta\beta'}\delta_{\delta\delta'}S_2^{\alpha\alpha'}\Big) + O(n_k\tau^{-3})\,.
\end{aligned}
$$

Note that the terms above can be computed using Lemma C.2.

$\square$

**Lemma C.5.**

$$\mathbb{E}_\ell \left[ A^{\alpha\alpha'} A^{\beta\beta'} A^{\delta\delta'} A^{\omega\omega'} \right] - \mathbb{E}_\ell \left[ A^{\alpha\alpha'} A^{\beta\beta'} \right] \mathbb{E}_\ell \left[ A^{\delta\delta'} A^{\omega\omega'} \right]$$
$$= \frac{n_k}{\tau^2 m^2} \left( \delta_{\alpha\alpha'} \delta_{\delta\delta'} S_1^{\beta\beta',\omega\omega'} + \delta_{\alpha\alpha'} \delta_{\omega\omega'} S_1^{\beta\beta',\delta\delta'} + \delta_{\beta\beta'} \delta_{\delta\delta'} S_1^{\alpha\alpha',\omega\omega'} + \delta_{\beta\beta'} \delta_{\omega\omega'} S_1^{\alpha\alpha',\delta\delta'} \right) + \mathcal{O}(n_k \tau^{-3}) \,.$$

*Proof.* The results is an immediate consequence of Lemma C.4 and Lemma C.3, where only the terms that do not cancel out are kept. $\square$

## C.4 Neural Covariance SDE for Stable Attention

Recall that we have the following model:

$$X_{\ell+1} = \lambda X_\ell + \gamma A_\ell X_\ell \frac{1}{\sqrt{n}} W_\ell^V$$

For the above model, $V_\ell$ has the following form:

$$V_{\ell+1} = \lambda^2 V_\ell + \frac{\lambda\gamma}{n\sqrt{n}} \left( X_\ell W_\ell^\top X_\ell^\top A_\ell^\top + A_\ell X_\ell W_\ell X_\ell^\top \right) + \frac{\gamma^2}{n^2} A_\ell X_\ell W_\ell W_\ell^\top X_\ell^\top A_\ell^\top \,.$$

We define:

$$\mathcal{T}_1^{\alpha\beta} := \frac{1}{n} \left( X_\ell W_\ell^\top X_\ell^\top A_\ell^\top + A_\ell X_\ell W_\ell X_\ell^\top \right)^{\alpha\beta} \,,$$
$$\mathcal{T}_2^{\alpha\beta} := \frac{1}{n\sqrt{n}} \left( A_\ell X_\ell W_\ell W_\ell^\top X_\ell^\top A_\ell^\top \right)^{\alpha\beta} \,.$$

Hence, the expression for $V_\ell$ simplifies to:

$$V_{\ell+1}^{\alpha\beta} = \lambda^2 V_\ell^{\alpha\beta} + \frac{\lambda\gamma}{\sqrt{n}} \mathcal{T}_1^{\alpha\beta} + \frac{\gamma^2}{\sqrt{n}} \mathcal{T}_2^{\alpha\beta} \,.$$

We need to compute the moments for these two quantities:

**Lemma C.6** (Moments of $\mathcal{T}_1$).

$$\mathbb{E}_\ell [\mathcal{T}_1^{\alpha\beta}] = 0 \,,$$

*and*

$$\mathbb{E}_\ell \left[ \mathcal{T}_1^{\alpha\beta} \mathcal{T}_1^{\delta\omega} \right] = 2(V^{\alpha\delta} V^{\beta\omega} + V^{\alpha\omega} V^{\beta\delta}) + \mathcal{O}(n_k \tau^{-3}) \,.$$

*Proof.*

$$\mathcal{T}_1^{\alpha\beta} \mathcal{T}_1^{\delta\omega} = \frac{1}{n^2} \left[ \left( X_\ell W_\ell^\top X_\ell^\top A_\ell^\top + A_\ell X_\ell W_\ell X_\ell^\top \right)^{\alpha\beta} \left( X_\ell W_\ell^\top X_\ell^\top A_\ell^\top + A_\ell X_\ell W_\ell X_\ell^\top \right)^{\delta\omega} \right]$$

$$= \frac{1}{n^2} \Bigg[ \left( X_\ell W_\ell^\top X_\ell^\top A_\ell^\top \right)^{\alpha\beta} \left( X_\ell W_\ell^\top X_\ell^\top A_\ell^\top \right)^{\delta\omega} + \left( X_\ell W_\ell^\top X_\ell^\top A_\ell^\top \right)^{\alpha\beta} \left( A_\ell X_\ell W_\ell X_\ell^\top \right)^{\delta\omega}$$

$$+ \left( A_\ell X_\ell W_\ell X_\ell^\top \right)^{\alpha\beta} \left( X_\ell W_\ell^\top X_\ell^\top A_\ell^\top \right)^{\delta\omega} + \left( A_\ell X_\ell W_\ell X_\ell^\top \right)^{\alpha\beta} \left( A_\ell X_\ell W_\ell X_\ell^\top \right)^{\delta\omega} \Bigg] \,.$$

Let's look at the first summand:

$$\frac{1}{n^2} \left( X_\ell W_\ell^\top X_\ell^\top A_\ell^\top \right)^{\alpha\beta} \left( X_\ell W_\ell^\top X_\ell^\top A_\ell^\top \right)^{\delta\omega} = \frac{1}{n^2} \sum_{\nu\kappa} \sum_{kk'jj'} X_k^\alpha W_{k'k} X_{k'}^\nu A^{\beta\nu} X_j^\delta W_{j'j} X_{j'}^\kappa A^{\omega\kappa} \,.$$

Hence, in expectation with respect to $W$:

$$\frac{1}{n^2}\sum_{\nu\kappa}\sum_{kk'jj'}X_k^\alpha\mathbb{E}\left[W_{k'k}W_{j'j}\right]X_{k'}^\nu A^{\beta\nu}X_j^\delta X_{j'}^\kappa A^{\omega\kappa} = \frac{1}{n^2}\sum_{\nu\kappa}\sum_{kk'jj'}X_k^\alpha\delta_{kj}\delta_{k'j'}X_{k'}^\nu A^{\beta\nu}X_j^\delta X_{j'}^\kappa A^{\omega\kappa}$$

$$= \frac{1}{n^2}\sum_{\nu\kappa}\sum_{kk'}X_k^\alpha X_{k'}^\nu A^{\beta\nu}X_k^\delta X_{k'}^\kappa A^{\omega\kappa}$$

$$= \sum_{\nu\kappa}V^{\alpha\delta}V^{\nu\kappa}A^{\beta\nu}A^{\omega\kappa}.$$

An identical argument can be made for the remaining three summands. Hence, taking expectation with respect to the Softmax weights:

$$\mathbb{E}_\ell\left[\mathcal{T}_1^{\alpha\beta}\mathcal{T}_1^{\delta\omega}\right] = \sum_{\nu\kappa}\left(V^{\alpha\delta}V^{\nu\kappa}\mathbb{E}_\ell[A^{\beta\nu}A^{\omega\kappa}] + V^{\alpha\omega}V^{\nu\kappa}\mathbb{E}_\ell[A^{\beta\nu}A^{\delta\kappa}] + V^{\beta\delta}V^{\nu\kappa}\mathbb{E}_\ell[A^{\alpha\nu}A^{\omega\kappa}] + V^{\beta\omega}V^{\nu\kappa}\mathbb{E}_\ell[A^{\alpha\nu}A^{\delta\kappa}]\right) .$$

Now, using Lemma C.3:

$$\mathbb{E}_\ell\left[\mathcal{T}_1^{\alpha\beta}\mathcal{T}_1^{\delta\omega}\right] = \sum_{\nu\kappa}V^{\nu\kappa}\left(V^{\alpha\delta}\mathbb{E}_\ell[A^{\beta\nu}A^{\omega\kappa}] + V^{\alpha\omega}\mathbb{E}_\ell[A^{\beta\nu}A^{\delta\kappa}] + V^{\beta\delta}\mathbb{E}_\ell[A^{\alpha\nu}A^{\omega\kappa}] + V^{\beta\omega}\mathbb{E}_\ell[A^{\alpha\nu}A^{\delta\kappa}]\right)$$

$$= \sum_{\nu\kappa}V^{\nu\kappa}\left(V^{\alpha\delta}\delta_{\beta\nu}\delta_{\omega\kappa} + V^{\alpha\omega}\delta_{\beta\nu}\delta_{\delta\kappa} + V^{\beta\delta}\delta_{\alpha\nu}\delta_{\omega\kappa} + V^{\beta\omega}\delta_{\alpha\nu}\delta_{\delta\kappa}\right) + \mathcal{O}(n_k\tau^{-2})$$

$$= V^{\beta\omega}V^{\alpha\delta} + V^{\alpha\omega}V^{\beta\delta} + V^{\beta\delta}V^{\alpha\omega} + V^{\beta\omega}V^{\alpha\delta} + \mathcal{O}(n_k\tau^{-2})$$

$$= 2(V^{\alpha\delta}V^{\beta\omega} + V^{\alpha\omega}V^{\beta\delta}) + \mathcal{O}(n_k\tau^{-2}).$$

$\square$

**Lemma C.7** (Moments of $\mathcal{T}_2$).

$$\mathbb{E}_\ell[\mathcal{T}_2^{\alpha\beta}] = \sqrt{n}\sum_{\nu\kappa}V^{\nu\kappa}\mathbb{E}_\ell\left[A^{\alpha\nu}A^{\beta\kappa}\right] ,$$

$$\mathbb{E}_\ell[\mathcal{T}_2^{\alpha\beta}\mathcal{T}_2^{\delta\omega}] = \sum_{\nu\kappa\nu'\kappa'}\mathbb{E}_\ell[A^{\alpha\nu}A^{\beta\kappa}A^{\delta\nu'}A^{\omega\kappa'}]\left(nV^{\nu\kappa}V^{\nu'\kappa'} + V^{\nu\nu'}V^{\kappa\kappa'} + V^{\nu\kappa'}V^{\nu'\kappa}\right) .$$

*Proof.*

$$\mathcal{T}_2^{\alpha\beta} := \frac{1}{n\sqrt{n}}\left(A_\ell X_\ell W_\ell W_\ell^\top X_\ell^\top A_\ell^\top\right)^{\alpha\beta} = \frac{1}{n\sqrt{n}}\sum_{\nu\kappa}\sum_{kk'j}A^{\alpha\nu}X_k^\nu W_{kk'}W_{jk'}X_j^\kappa A^{\beta\kappa}.$$

Taking expectation with respect to $W$:

$$\frac{1}{n\sqrt{n}}\left(A_\ell X_\ell W_\ell W_\ell^\top X_\ell^\top A_\ell^\top\right)^{\alpha\beta} = \frac{1}{n\sqrt{n}}\sum_{\nu\kappa}\sum_{kk'j}A^{\alpha\nu}X_k^\nu\mathbb{E}[W_{kk'}W_{jk'}]X_j^\kappa A^{\beta\kappa}$$

$$= \frac{1}{n\sqrt{n}}\sum_{\nu\kappa}\sum_{kk'j}A^{\alpha\nu}X_k^\nu\delta_{kj}X_j^\kappa A^{\beta\kappa}$$

$$= \frac{1}{n\sqrt{n}}\sum_{\nu\kappa}\sum_{kk'}A^{\alpha\nu}A^{\beta\kappa}X_k^\nu X_k^\kappa$$

$$= \frac{1}{\sqrt{n}}\sum_{\nu\kappa}\sum_{k}A^{\alpha\nu}A^{\beta\kappa}X_k^\nu X_k^\kappa$$

$$= \sqrt{n}\sum_{\nu\kappa}V^{\nu\kappa}A^{\alpha\nu}A^{\beta\kappa}.$$

Taking expectation w.r.t the Softmax weights, we get the desired result.

For second moment, we can take the conditional expectation:

$$\mathbb{E}_\ell[\mathcal{T}_2^{\alpha\beta}\mathcal{T}_2^{\delta\omega}] = \frac{1}{n^3}\sum_{\nu\kappa\nu'\kappa'}\sum_{kk'jii'j'}A^{\alpha\nu}A^{\beta\kappa}A^{\delta\nu'}A^{\omega\kappa'}X_k^\nu X_j^\kappa X_i^{\nu'}X_{j'}^{\kappa'}\mathbb{E}[W_{kk'}W_{jk'}W_{ii'}W_{j'i'}],$$

where we recall $\mathbb{E}_\ell[\cdot] = \mathbb{E}[\cdot|\mathcal{F}_\ell]$ is the conditional expectation given the sigma-algebra generated by $\mathcal{F}_\ell = \sigma(\{X_k\}_{k\in[\ell]})$.

Using Isserlis Theorem, we have that:

$$\mathbb{E}[W_{kk'}W_{jk'}W_{ii'}W_{j'i'}] = \delta_{kj}\delta_{ij'} + \delta_{ki}\delta_{k'i'}\delta_{jj'} + \delta_{kj'}\delta_{k'i'}\delta_{ji}.$$

Hence:

$$\mathbb{E}_\ell[\mathcal{T}_2^{\alpha\beta}\mathcal{T}_2^{\delta\omega}] = \frac{1}{n^3}\sum_{\nu\kappa\nu'\kappa'}A^{\alpha\nu}A^{\beta\kappa}A^{\delta\nu'}A^{\omega\kappa'}\sum_{kk'jii'j'}X_k^\nu X_j^\kappa X_i^{\nu'}X_{j'}^{\kappa'}\Big(\delta_{kj}\delta_{ij'} + \delta_{ki}\delta_{k'i'}\delta_{jj'} +$$

$$+ \delta_{kj'}\delta_{k'i'}\delta_{ji}\Big)$$

$$= \frac{1}{n^3}\sum_{\nu\kappa\nu'\kappa'}A^{\alpha\nu}A^{\beta\kappa}A^{\delta\nu'}A^{\omega\kappa'}\Big(\sum_{kk'ii'}X_k^\nu X_k^\kappa X_i^{\nu'}X_i^{\kappa'} + \sum_{kk'j}X_k^\nu X_j^\kappa X_k^{\nu'}X_j^{\kappa'}$$

$$+ \sum_{kk'j}X_k^\nu X_j^\kappa X_j^{\nu'}X_k^{\kappa'}\Big)$$

$$= \frac{1}{n^3}\sum_{\nu\kappa\nu'\kappa'}A^{\alpha\nu}A^{\beta\kappa}A^{\delta\nu'}A^{\omega\kappa'}\Big(n^4 V^{\nu\kappa}V^{\nu'\kappa'} + n^3 V^{\nu\nu'}V^{\kappa\kappa'} + n^3 V^{\nu\kappa'}V^{\nu'\kappa}\Big)$$

$$= \sum_{\nu\kappa\nu'\kappa'}A^{\alpha\nu}A^{\beta\kappa}A^{\delta\nu'}A^{\omega\kappa'}\Big(nV^{\nu\kappa}V^{\nu'\kappa'} + V^{\nu\nu'}V^{\kappa\kappa'} + V^{\nu\kappa'}V^{\nu'\kappa}\Big).$$

By taking expectation w.r.t the Softmax parameters, we get the desired result. □

**Lemma C.8** (Covariance of $\mathcal{T}_2$).

$$\mathbb{E}_\ell[\mathcal{T}_2^{\alpha\beta}\mathcal{T}_2^{\delta\omega}] - \mathbb{E}_\ell[\mathcal{T}_2^{\alpha\beta}]\mathbb{E}_\ell[\mathcal{T}_2^{\delta\omega}] = \frac{nn_k}{\tau^2}\mathcal{A}^{\alpha\beta\delta\omega} + V^{\alpha\delta}V^{\beta\omega} + V^{\alpha\omega}V^{\beta\delta} + \mathcal{O}\left(\frac{nn_k}{\tau^3} + \frac{n_k}{\tau^2}\right),$$

*where:*

$$\mathcal{A}^{\alpha\beta\delta\omega} := \frac{1}{m^2}\sum_{\nu\kappa}\left(V^{\alpha\kappa}V^{\delta\nu}S_1^{\beta\kappa,\omega\nu} + V^{\alpha\kappa}V^{\omega\nu}S_1^{\beta\kappa,\delta\nu} + V^{\beta\nu}V^{\delta\kappa}S_1^{\alpha\nu,\omega\kappa} + V^{\beta\nu}V^{\omega\kappa}S_1^{\alpha\nu,\delta\kappa}\right).$$

*Proof.* Using Lemma C.7, we have that:

$$\mathbb{E}_\ell[\mathcal{T}_2^{\alpha\beta}\mathcal{T}_2^{\delta\omega}] - \mathbb{E}_\ell[\mathcal{T}_2^{\alpha\beta}]\mathbb{E}_\ell[\mathcal{T}_2^{\delta\omega}] = \sum_{\nu\kappa\nu'\kappa'}\mathbb{E}_\ell[A^{\alpha\nu}A^{\beta\kappa}A^{\delta\nu'}A^{\omega\kappa'}]\left(nV^{\nu\kappa}V^{\nu'\kappa'} + V^{\nu\nu'}V^{\kappa\kappa'} + V^{\nu\kappa'}V^{\nu'\kappa}\right)$$

$$- n\sum_{\nu\kappa\nu'\kappa'}V^{\nu\kappa}V^{\nu'\kappa'}\mathbb{E}_\ell\left[A^{\alpha\nu}A^{\beta\kappa}\right]\mathbb{E}_\ell\left[A^{\delta\nu'}A^{\omega\kappa'}\right]$$

$$= n\sum_{\nu\kappa\nu'\kappa'}V^{\nu\kappa}V^{\nu'\kappa'}\left(\mathbb{E}_\ell[A^{\alpha\nu}A^{\beta\kappa}A^{\delta\nu'}A^{\omega\kappa'}] - \mathbb{E}_\ell\left[A^{\alpha\nu}A^{\beta\kappa}\right]\mathbb{E}_\ell\left[A^{\delta\nu'}A^{\omega\kappa'}\right]\right)$$

$$+ \sum_{\nu\kappa\nu'\kappa'}\mathbb{E}_\ell[A^{\alpha\nu}A^{\beta\kappa}A^{\delta\nu'}A^{\omega\kappa'}]\left(V^{\nu\nu'}V^{\kappa\kappa'} + V^{\nu\kappa'}V^{\nu'\kappa}\right).$$

Now we can use Lemma C.3 and Lemma C.4 to compute the moments of $A$. For the second summand, we simply have:

$$\mathbb{E}_\ell\left[A^{\alpha\nu}A^{\beta\kappa}A^{\delta\nu'}A^{\omega\kappa'}\right] = \delta_{\alpha\nu}\delta_{\beta\kappa}\delta_{\delta\nu'}\delta_{\omega\kappa'} + \mathcal{O}(n_k\tau^{-2}),$$

hence:

$$\sum_{\nu\kappa\nu'\kappa'}\mathbb{E}_\ell[A^{\alpha\nu}A^{\beta\kappa}A^{\delta\nu'}A^{\omega\kappa'}]\left(V^{\nu\nu'}V^{\kappa\kappa'} + V^{\nu\kappa'}V^{\nu'\kappa}\right) = V^{\alpha\delta}V^{\beta\omega} + V^{\alpha\omega}V^{\beta\delta} + \mathcal{O}(n_k\tau^{-2}).$$

For the first summand, recall from Lemma C.5 that:

$$\mathbb{E}_\ell\left[A^{\alpha\alpha'}A^{\beta\beta'}A^{\delta\delta'}A^{\omega\omega'}\right] - \mathbb{E}_\ell\left[A^{\alpha\alpha'}A^{\beta\beta'}\right]\mathbb{E}_\ell\left[A^{\delta\delta'}A^{\omega\omega'}\right]$$

$$= \frac{n_k}{\tau^2 m^2}\left(\delta_{\alpha\alpha'}\delta_{\delta\delta'}S_1^{\beta\beta',\omega\omega'} + \delta_{\alpha\alpha'}\delta_{\omega\omega'}S_1^{\beta\beta',\delta\delta'} + \delta_{\beta\beta'}\delta_{\delta\delta'}S_1^{\alpha\alpha',\omega\omega'} + \delta_{\beta\beta'}\delta_{\omega\omega'}S_1^{\alpha\alpha',\delta\delta'}\right)$$

$$+ \mathcal{O}(n_k\tau^{-3}).$$

Hence:

$$n \sum_{\nu\kappa\nu'\kappa'} V^{\nu\kappa}V^{\nu'\kappa'} \left( \mathbb{E}_\ell[A^{\alpha\nu}A^{\beta\kappa}A^{\delta\nu'}A^{\omega\kappa'}] - \mathbb{E}_\ell\left[A^{\alpha\nu}A^{\beta\kappa}\right]\mathbb{E}_\ell\left[A^{\delta\nu'}A^{\omega\kappa'}\right] \right)$$

$$= \frac{nn_k}{\tau^2 m^2} \sum_{\nu\kappa\nu'\kappa'} V^{\nu\kappa}V^{\nu'\kappa'} \left( \delta_{\alpha\nu}\delta_{\delta\nu'}S_1^{\beta\kappa,\omega\kappa'} + \delta_{\alpha\nu}\delta_{\omega\kappa'}S_1^{\beta\kappa,\delta\nu'} + \delta_{\beta\kappa}\delta_{\delta\nu'}S_1^{\alpha\nu,\omega\kappa'} + \delta_{\beta\kappa}\delta_{\omega\kappa'}S_1^{\alpha\nu,\delta\nu'} \right)$$

$$+ \mathcal{O}(nn_k\tau^{-3})$$

$$= \frac{nn_k}{\tau^2}\frac{1}{m^2} \underbrace{\sum_{\nu\kappa} \left( V^{\alpha\kappa}V^{\delta\nu}S_1^{\beta\kappa,\omega\nu} + V^{\alpha\kappa}V^{\omega\nu}S_1^{\beta\kappa,\delta\nu} + V^{\beta\nu}V^{\delta\kappa}S_1^{\alpha\nu,\omega\kappa} + V^{\beta\nu}V^{\omega\kappa}S_1^{\alpha\nu,\delta\kappa} \right)}_{\mathcal{A}^{\alpha\beta\delta\omega}}$$

$$+ \mathcal{O}(nn_k\tau^{-3}).$$

$\square$

We are now ready to re-state and proof of Theorem 4.2.

**Theorem 4.2.** *Let $X_\ell$ be the hidden layers of a residual attention network defined in Eq. 1 with shaped attention in Eq. 9, parameters $\lambda^2 + \gamma^2 = 1$ and $\tau = \tau_0\sqrt{nn_k}$, where $\lambda, \gamma, \tau_0$ all do not depend on $d, n$. Then the feature covariance $V_\ell$ converges locally to the solution of the following SDE (in the sense of Definition 4.1)*

$$dV_t = b(V_t)dt + \Sigma(V_t)^{1/2}dB_t, \quad V_0 = \frac{1}{n}X_0X_0^\top,$$

*where the drift has the following form*

$$b(V) = \frac{\gamma^2}{\tau_0^2} \left[ \frac{1}{m^2}\sum_{\nu,\kappa=1}^m V^{\nu\kappa}S_1^{\alpha\nu,\beta\kappa} + \frac{1}{2m}\sum_{\nu=1}^m(V^{\beta\nu}S_2^{\alpha\nu} + V^{\alpha\nu}S_2^{\beta\nu}) \right]_{\alpha\leq\beta},$$

*the diffusion coefficient is defined by $\Sigma(V) = \gamma^2(2-\gamma^2)\Sigma_{lin}(V) + \gamma^4\tau_0^{-2}[\mathcal{A}^{\alpha\beta,\delta\omega}]_{\alpha\leq\beta,\delta\leq\omega}$, and*

$$\mathcal{A}^{\alpha\beta,\delta\omega} := \frac{1}{m^2}\sum_{\nu,\kappa=1}^m \left( V^{\alpha\kappa}V^{\delta\nu}S_1^{\beta\kappa,\omega\nu} + V^{\alpha\kappa}V^{\omega\nu}S_1^{\beta\kappa,\delta\nu} + V^{\beta\nu}V^{\delta\kappa}S_1^{\alpha\nu,\omega\kappa} + V^{\beta\nu}V^{\omega\kappa}S_1^{\alpha\nu,\delta\kappa} \right).$$

*Proof.* Recall that:

$$V_{\ell+1}^{\alpha\beta} = \lambda^2 V_\ell^{\alpha\beta} + \frac{\lambda\gamma}{\sqrt{n}}\mathcal{T}_1^{\alpha\beta} + \frac{\gamma^2}{\sqrt{n}}\mathcal{T}_2^{\alpha\beta}.$$

**Drift.** Summing and subtracting $\mathbb{E}_\ell[\mathcal{T}_2]$ (and using Lemma C.7), we have that:

$$V_{\ell+1}^{\alpha\beta} = \lambda^2 V_\ell^{\alpha\beta} + \gamma^2\sum_{\nu\kappa}V^{\nu\kappa}\mathbb{E}_\ell\left[A^{\alpha\nu}A^{\beta\kappa}\right] + \frac{\lambda\gamma}{\sqrt{n}}\mathcal{T}_1^{\alpha\beta} + \frac{\gamma^2}{\sqrt{n}}\left(\mathcal{T}_2^{\alpha\beta} - \mathbb{E}_\ell[\mathcal{T}_2]\right),$$

where we recall $\mathbb{E}_\ell[\,\cdot\,] = \mathbb{E}[\,\cdot\,|\mathcal{F}_\ell]$ is the conditional expectation given the sigma-algebra generated by $\mathcal{F}_\ell = \sigma(\{X_k\}_{k\in[\ell]})$.

Re-stating Lemma C.3, we have that:

$$\mathbb{E}_\ell[A^{\alpha\nu}A^{\beta\kappa}] = \delta_{\alpha\nu}\delta_{\beta\kappa} + \frac{n_k}{\tau^2 m^2}S_1^{\alpha\nu,\beta\kappa} + \frac{n_k}{2\tau^2 m}(\delta_{\beta\kappa}S_2^{\alpha\nu} + \delta_{\alpha\nu}S_2^{\beta\kappa}) + O(n_k\tau^{-3}).$$

Plugging in the expression, and using $\lambda^2 + \gamma^2 = 1$, we have that the drift is:

$$\lambda^2 V_\ell^{\alpha\beta} + \gamma^2 V_\ell^{\alpha\beta} + \gamma^2\frac{n_k}{\tau^2 m}\sum_{\nu\kappa}V^{\nu\kappa}\left(\frac{1}{m}S_1^{\alpha\nu,\beta\kappa} + \frac{1}{2}(\delta_{\beta\kappa}S_2^{\alpha\nu} + \delta_{\alpha\nu}S_2^{\beta\kappa})\right) + O(n_k\tau^{-3})$$

$$= V_\ell^{\alpha\beta} + \gamma^2\frac{n_k}{\tau^2}\left(\frac{1}{m^2}\sum_{\nu\kappa}V^{\nu\kappa}S_1^{\alpha\nu,\beta\kappa} + \frac{1}{2m}\sum_\nu(V^{\beta\nu}S_2^{\alpha\nu} + V^{\alpha\nu}S_2^{\beta\nu})\right) + O(n_k\tau^{-3}).$$

From here, it is evident that in order to have the drift scaling as $\mathcal{O}(1/n)$ we need to choose:

$$\tau^2 = \tau_0^2 n n_k, \tag{26}$$

where $\tau_0 > 0$ is a constant.

**Covariance.** Recall that:

$$V_{\ell+1}^{\alpha\beta} = \lambda^2 V_\ell^{\alpha\beta} + \gamma^2 \sum_{\nu\kappa} V^{\nu\kappa} \mathbb{E}_\ell \left[ A^{\alpha\nu} A^{\beta\kappa} \right] + \frac{\lambda\gamma}{\sqrt{n}} \mathcal{T}_1^{\alpha\beta} + \frac{\gamma^2}{\sqrt{n}} \left( \mathcal{T}_2^{\alpha\beta} - \mathbb{E}_\ell[\mathcal{T}_2] \right).$$

Furthermore, we have set $\lambda^2 + \gamma^2 = 1$ and $\tau^2 = \tau_0^2 n n_k$ to have the right scaling for the drift.

What's left to is to compute the conditional covariance for $V_{\ell+1}$ given $\mathcal{F}_\ell$. Noting that $\mathcal{T}_1^{\alpha\beta}, \mathcal{T}_2^{\delta\omega}$ are uncorrelated, i.e. $\mathbb{E}_\ell \left[ \mathcal{T}_1^{\alpha\beta} \mathcal{T}_2^{\delta\omega} \right] = 0$ (similarly to the case of Resnet with shaped ReLU Lemma B.3), we have that:

$$\text{Cov}_\ell \left( V_{\ell+1}^{\alpha\beta}, V_{\ell+1}^{\delta\omega} \right) = \mathbb{E}_\ell \left[ \left( \lambda\gamma \mathcal{T}_1^{\alpha\beta} + \gamma^2 \mathcal{T}_2^{\alpha\beta} - \gamma^2 \mathbb{E}_\ell[\mathcal{T}_2^{\alpha\beta}] \right) \left( \lambda\gamma \mathcal{T}_1^{\delta\omega} + \gamma^2 \mathcal{T}_2^{\delta\omega} - \gamma^2 \mathbb{E}_\ell[\mathcal{T}_2^{\delta\omega}] \right) \right]$$

$$= \lambda^2 \gamma^2 \mathbb{E}_\ell \left[ \mathcal{T}_1^{\alpha\beta} \mathcal{T}_1^{\delta\omega} \right] + \gamma^4 \mathbb{E}_\ell \left[ \mathcal{T}_2^{\alpha\beta} \mathcal{T}_2^{\delta\omega} \right] - \gamma^4 \mathbb{E}_\ell \left[ \mathcal{T}_2^{\alpha\beta} \right] \mathbb{E}_\ell \left[ \mathcal{T}_2^{\delta\omega} \right],$$

where we use $\text{Cov}_\ell$ to denote the conditional covariance given the sigma-algebra generated by $\mathcal{F}_\ell = \sigma(\{X_k\}_{k\in[\ell]})$.

Using Lemma C.6 and Lemma C.8, we have that:

$$\text{Cov}_\ell \left( V_{\ell+1}^{\alpha\beta}, V_{\ell+1}^{\delta\omega} \right) = \lambda^2 \gamma^2 \left( 2(V^{\alpha\delta} V^{\beta\omega} + V^{\alpha\omega} V^{\beta\delta}) \right) + \gamma^4 \left( \frac{1}{\tau_0^2} \mathcal{A}^{\alpha\beta\delta\omega} + V^{\alpha\delta} V^{\beta\omega} + V^{\alpha\omega} V^{\beta\delta} \right)$$

$$= \gamma^2 (2 - \gamma^2) \left( V^{\alpha\delta} V^{\beta\omega} + V^{\alpha\omega} V^{\beta\delta} \right) + \frac{\gamma^4}{\tau_0^2} \mathcal{A}^{\alpha\beta\delta\omega}.$$

Now we can apply Proposition A.2 for locally Lipschitz drift and covariance coefficients, which gives us the desired result in local convergence in the Skorohod topology.

$\square$

We will also restate and prove Corollary 4.3.

**Corollary 4.3** (Shaped Transformer Covariance SDE). *Let $X_\ell$ be the hidden layers of a shaped transformer defined in Eq. 11 with parameters $\lambda^2 + \gamma^2 = 1$ and $\tau = \tau_0 \sqrt{n n_k}$, where $\lambda, \gamma, \tau_0$ all do not depend on $d, n$. Then the feature covariance $V_\ell$ converges locally to the solution of the following SDE (in the sense of Definition 4.1)*

$$dV_t = [b(V_t) + b_{res}(V_t)] \, dt + [\Sigma(V_t) + \Sigma_{res}(V_t)]^{1/2} \, dB_t, \tag{12}$$

*where the coefficients are defined in Theorem 3.2 and Theorem 4.2.*

*Proof.* To combine the results of Theorem 3.2 and Theorem 4.2, it is sufficient to combine the following (simplified) iterated Markov updates into one Markov chain

$$U_\ell = V_\ell + \frac{b(V_\ell)}{n} + \frac{\Sigma(V_\ell)^{1/2} \xi_\ell}{\sqrt{n}}, \quad V_{\ell+1} = U_\ell + \frac{b_{\text{ReLU}}(U_\ell)}{n} + \frac{\Sigma_{\text{lin}}(U_\ell)^{1/2} \xi'_\ell}{\sqrt{n}}, \tag{27}$$

where $\xi_\ell, \xi'_\ell$ are independent zero mean and identity covariance random vectors.

Since in the limit, we have that either updates are infinitesimal, i.e.

$$|U_\ell - V_\ell| \xrightarrow{n\to\infty} 0 \quad \text{almost surely}, \tag{28}$$

then we can write $b_{\text{ReLU}(U_\ell)} = \widehat{b}_{\text{ReLU},n}(V_\ell, \omega_\ell)$, where eventually we have that

$$\lim_{n\to\infty} \mathbb{E}_\ell \left[ \widehat{b}_n(V_\ell, \omega_\ell) \right] = b(V_\ell), \tag{29}$$

so it will not contribute towards affecting the limit. Here we recall $\mathbb{E}_\ell[\,\cdot\,] = \mathbb{E}[\,\cdot\,|\mathcal{F}_\ell]$ is the conditional expectation given the sigma-algebra generated by $\mathcal{F}_\ell = \sigma(\{X_k\}_{k\in[\ell]})$. We can treat $\Sigma_{\text{lin}}(U_\ell)$ similarly to get the Markov chain update

$$V_{\ell+1} = V_\ell + \frac{b(V_\ell) + \widehat{b}_{\text{ReLU},n}(V_\ell, \omega_\ell)}{n} + \frac{\Sigma(V_\ell)^{1/2}\xi_\ell}{\sqrt{n}} + \frac{\widehat{\Sigma}_{\text{lin},n}(V_\ell)^{1/2}\xi_\ell'}{\sqrt{n}}\,, \qquad (30)$$

which converges to the following SDE with two Brownian motions using Proposition A.2

$$dV_t = [b(V_t) + b_{\text{res}}(V_t)]\,dt + \Sigma(V_t)^{1/2}\,dB_t + \Sigma_{\text{res}}(V_t)^{1/2}dB_t'\,. \qquad (31)$$

Observe that since the two Brownian motions $B_t, B_t'$ are independent, it's equivalent to write

$$\Sigma(V_t)^{1/2}\,dB_t + \Sigma_{\text{res}}(V_t)^{1/2}dB_t' \overset{d}{=} [\Sigma(V_t) + \Sigma_{\text{res}}(V_t)^{1/2}]^{1/2}\,dB_t\,. \qquad (32)$$

We recover the desired results from considering a more general form of the iterated Markov updates as in Proposition A.2, which do not hinder the above derivation.

$\square$

# D    Preliminary Experiments

We perform preliminary experiments to understand the effect of *shaped attention* on training deep Transformer architectures. In particular, we consider a pre-training masked language modeling task, where we mask 15% of the tokens. We use a subset of the English Wikipedia *20220301.en* and English *bookcorpus* datasets [69, 70]. As a baseline, we adopt a Pre-LN Transformer encoder architecture with 18 or 24 blocks. For the residual feedforward layer, we shape the ReLU activation according to Eq. 4, by changing its negative slope to $s_- = 1 - 1/\sqrt{n}$ instead of 0. We then incorporate our shaped attention by replacing the attention mechanism as dictated in Eq. 9. We also add scalar multipliers $\gamma_1, \gamma_2 \in \mathbb{R}$ both to the identity and centering terms of the shaped attention:

$$A_\ell = \gamma_1 I + \text{Softmax}(\tau^{-1}Y_\ell) - \gamma_2 \frac{1}{m}\mathbf{1}\mathbf{1}^\top\,, \quad \tau = \tau_0\sqrt{nn_k}\,, \qquad (33)$$

and propose two ways to set $\gamma_1, \gamma_2$ during training. In both alternatives we initialize $\gamma_1, \gamma_2 = 1$, thus leveraging the stability properties of shaped attention at initialization. During training, we either:

1. **Recover**. Linearly decrease $\gamma_1, \gamma_2$ to zero with in the first 4000 steps, thus recovering the standard attention layer. Apply the same schedule for the shaped-ReLU slope $s_-$, recovering the usual ReLU activation. This approach recovers the vanilla Transformer architecture (without LayerNorm).

2. **Learn**. Learn all the shaping parameters $\gamma_1, \gamma_2$ and $s_-$.

The intuitive rationale behind these choices is that at initialization we want good signal propagation and a non-degenerate covariance (according to our theory, this requires the shaped attention and ReLU). On the other hand, we also allow the model to more dynamically make use of the nonlinearity during training to modify the correlation structure with **Recover** or **Learn**. We report that without either adjustment, *shaped attention* is still trainable but at much slower rates.

We also incorporate the $\frac{1}{\sqrt{n}}$ factor into the initialization of the queries and keys weights by decreasing the variance of their entries by a factor of $\frac{1}{n}$. This allows us to not re-tune the learning rate for the queries and keys. We stress that at at initialization, the two formulations ($\tau = \sqrt{nn_k}$ and $\tau = \sqrt{n_k}$ with decreased weight's variance) are equivalent. In both alternatives we train all the skip connection parameters $\lambda$, $\gamma$, and initialize $\gamma$ in the grid $(0.05, 0.1, 0.2)$ and set $\tau_0 = 1$. We report training instabilities (loss divergence) for larger values of $\gamma$. All models —including the baselines — use Adam [71] with learning rate warmup of 4000 steps, the learning rate is tuned in the grid $(0.0001, 0.0005, 0.001, 0.005)$. We report the train/test loss after $100K$ optimization steps, averaged over 4 random seeds. All the other experimental details can be found in Appendix D.1.

**The Shaped Transformer is Trainable.** In Table 1, we compare the train/test loss of the two variant of shaped attention with the baseline Pre-LN model. Notice that our model (in both variants) achieves comparable performance to standard Transformers across all the reported values of $\gamma$.

**GLUE evaluation** Furthermore, we evaluate the trained models on three datasets from the GLUE benchmark [72] and summarize the results in Table 2. These demonstrate that our shaped Transformers holds promise by outperforming our pre-ln baseline.

|  | $\gamma$ | $d = 18$ | | $d = 24$ | |
|---|---|---|---|---|---|
|  |  | Train Loss | Test Loss | Train Loss | Test Loss |
| Learn | 0.05 | $2.09_{\pm 0.02}$ | $2.07_{\pm 0.02}$ | $2.03_{\pm 0.02}$ | $2.03_{\pm 0.01}$ |
|  | 0.1 | $2.10_{\pm 0.02}$ | $2.07_{\pm 0.01}$ | $\mathbf{2.02}_{\pm 0.01}$ | $2.02_{\pm 0.03}$ |
|  | 0.2 | $2.07_{\pm 0.05}$ | $2.06_{\pm 0.02}$ | $2.09_{\pm 0.04}$ | $2.09_{\pm 0.03}$ |
| Recover | 0.05 | $2.05_{\pm 0.03}$ | $2.04_{\pm 0.02}$ | $2.02_{\pm 0.04}$ | $2.00_{\pm 0.01}$ |
|  | 0.1 | $2.05_{\pm 0.01}$ | $2.04_{\pm 0.01}$ | $2.06_{\pm 0.03}$ | $2.00_{\pm 0.02}$ |
|  | 0.2 | $\mathbf{2.01}_{\pm 0.03}$ | $\mathbf{2.03}_{\pm 0.01}$ | $2.05_{\pm 0.04}$ | $2.00_{\pm 0.01}$ |
| Baseline |  | $2.08_{\pm 0.03}$ | $2.07_{\pm 0.01}$ | $2.08_{\pm 0.03}$ | $2.01_{\pm 0.01}$ |

Table 1: Training and test loss of the proposed *shaped attention*, both in the "Recover" and "Learn" alternatives. Baseline refers to the vanilla Pre-LN Transformer. We report the best run under the learning rates $(0.0001, 0.0005, 0.001, 0.005)$, averaged over 4 random seeds. We include the confidence interval of $\pm$ one standard deviation.

|  | Model | COLA | MRPC | RTE |
|---|---|---|---|---|
| $d = 18$ | Baseline | $0.139_{\pm 0.008}$ | $0.797_{\pm 0.010}$ | $0.504_{\pm 0.024}$ |
|  | Learn, $\gamma = 0.2$ | $0.182_{\pm 0.041}$ | $0.813_{\pm 0.005}$ | $0.528_{\pm 0.019}$ |
|  | Recover, $\gamma = 0.2$ | $0.221_{\pm 0.042}$ | $0.809_{\pm 0.009}$ | $0.520_{\pm 0.029}$ |
| $d = 24$ | Baseline | $0.022_{\pm 0.055}$ | $0.785_{\pm 0.009}$ | $0.48_{\pm 0.046}$ |
|  | Learn, $\gamma = 0.05$ | $0.150_{\pm 0.026}$ | $0.812_{\pm 0.004}$ | $0.513_{\pm 0.019}$ |
|  | Recover, $\gamma = 0.05$ | $0.211_{\pm 0.039}$ | $0.803_{\pm 0.012}$ | $0.529_{\pm 0.019}$ |
|  | BERT (Geiping et al.) | 0.103 | 74.8 | 0.509 |

Table 2: Evaluation of the pretrained models on a subset of the GLUE benchmark. We take the checkpoints of the baselines and selected shaped Transformers ($\gamma = 0.2$ for the shallower model $d = 18$, $\gamma = 0.05$ for the deeper model $d = 24$). Then, we fine-tune on each of three datasets from GLUE (COLA, MRPC, RTE) for 5 epochs and Adam optimizer $lr = 5e - 4$ with batch size 16, and report test evaluation metric for the corresponding task.

**Entropy Collapse for Large Learning rates.** To understand the sources of training instability, we keep track of the entropy of the probability distribution induced by the Softmax, as it has been observed that the Transformer's training is unstable in the low-entropy regime [57]. The entropy is calculated for each row of the Softmax matrix, and it is averaged across rows and heads. The results are in Fig. 6. Notice how for the large learning rate regime observed in Fig. **??**, the entropy collapses for the baseline model, but not for the proposed *shaped Transformer*. Entropy collapse indicates that the Softmax distribution degenerates to a point mass, which is itself caused by large logits. Remarkably, this phenomenon does not affect the `recover` setting, despite recovering the Transformer architecture (without layer normalization) after the warm-up period.

### D.1 Experimental details

**Dataset.** We use a subset of the English Wikipedia *20220301.en* and English *bookcorpus* datasets [69, 70]. The sentences are tokenized using by pre-training a tokenizer on the training set. We use a vocabulary size of 32000, and a maximum sequence length of 128 tokens.

**Model parameters.** We use an embedding size of $n = 768$ and 8 multi-attention heads. The batch size is fixed to 32 sequences. All the initial weights are sampled from $\mathcal{N}(0, n^{-1})$, with the exception of the queries and keys' weights $W^K, W^Q$ in the *shaped attention* case, that are sampled from $\mathcal{N}(0, n^{-3/2})$ (as explained in Appendix D). The feedforward layer maps the $n = 768$-dimensional embedding to the larger dimension 3072, as in the Hugging face implementation of Bert [73].

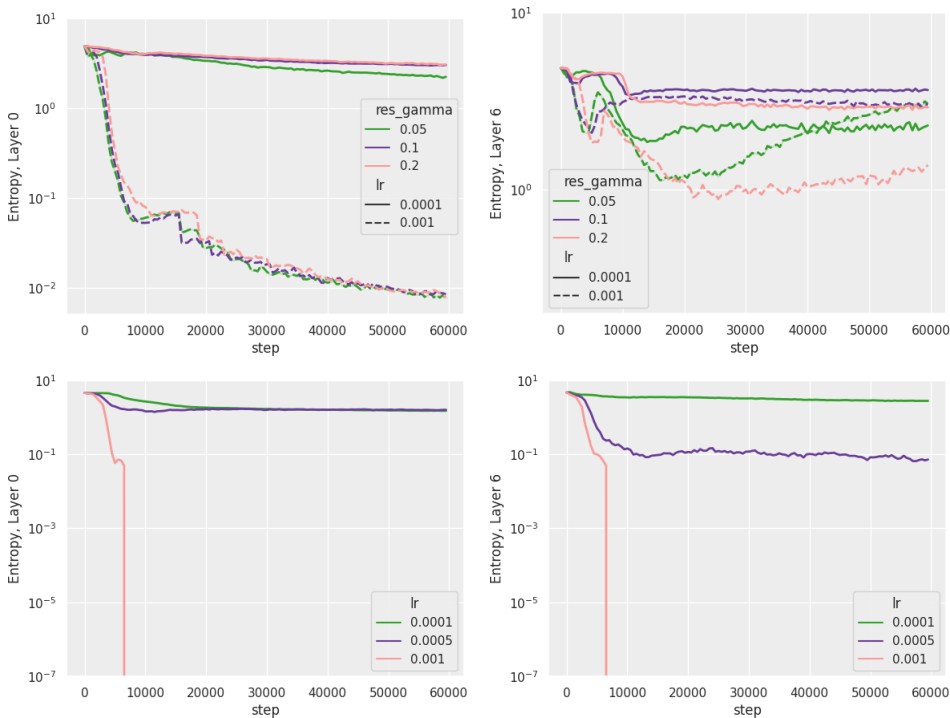

Figure 6: Dynamics of the mean entropy across heads for selected layers (first and seventh) and learning rates. (Above): *shaped attention* in `recover` setting. (Below): Baseline transformers. Notice that at $lr = 0.001$ the entropy collapses for the baseline model.

**Optimization.** We train using Adam [71] with betas parameters $(0.9, 0.999)$ and learning rate chosen in the grid $(0.0001, 0.0005, 0.001, 0.005)$. We do not use weight decay.

**Computational Resources.** The experiments are executed on Nvidia DGX-1 GPU nodes equipped with 4 20-core Xeon E5-2698v4 processors, 512 GB of memory and 8 Nvidia V100 GPUs.

# E   Additional Figures

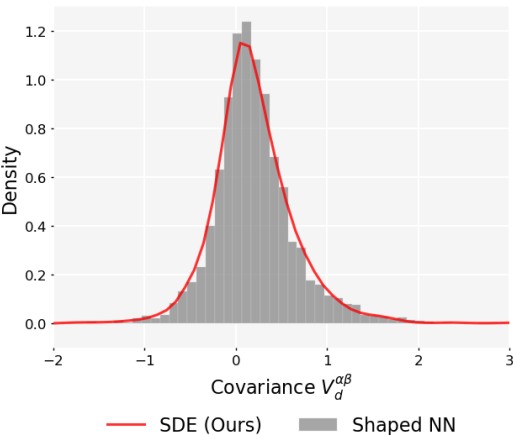

Figure 7: Kernel density estimate and histogram of covariances from the covariance SDE in Theorem 4.2 and shaped attention NN (Eq. 9). Simulated with $n = 200, d = 150, \gamma = 1/\sqrt{8}, \tau_0 = 1, V_0^{\alpha\beta} = 0.2$, SDE step size 0.01, and $2^{12}$ samples.

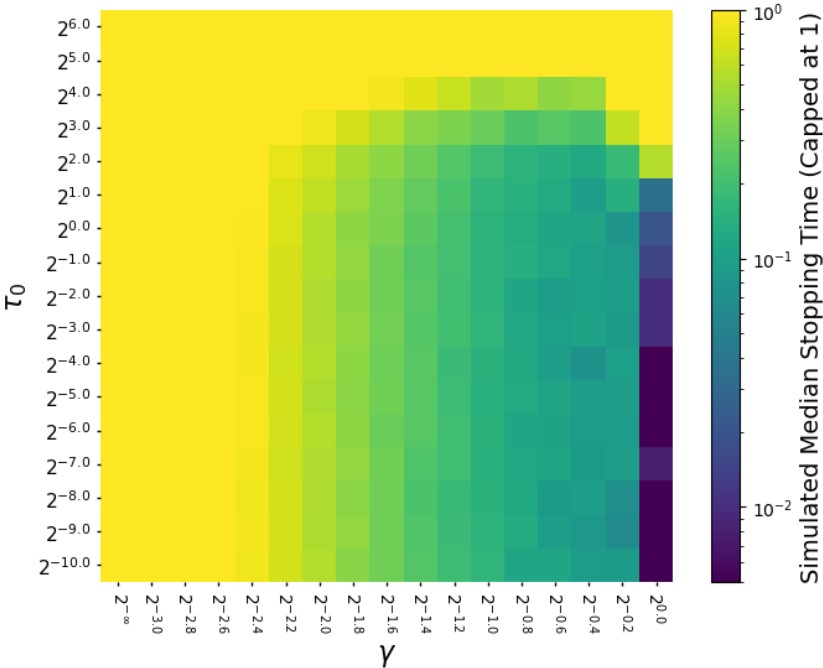

Figure 8: Median of the stopping time capped at 1, of the shaped attention neural network with respect to its parameters $\gamma$ and $\tau_0$. Stopping time is defined as $t^* = d^*/n$ with $d^*$ the maximum depth beyond which one of the eigenvalues of the covariance matrix exceeds $10^4$ or drops below $10^{-4}$. Simulated with $n = d = 200$, and 100 samples used to estimate the median. To demonstrate the potential numerical instabilities, we had to choose an *adversarial* set of parameters: in particular, an unrealistically large norm (approx. $10\sqrt{n}$) for the initial tokens $X_0$, which enlarges the eigenvalues of $V_0$ to the order of 100.

