# OpenReview forum: "The Shaped Transformer: Attention Models in the Infinite Depth-and-Width Limit"
_NeurIPS.cc/2023/Conference — NeurIPS 2023 poster_

### Official Review · Reviewer_h41x · 2023-07-03

**Soundness:** 3 good
**Presentation:** 3 good
**Contribution:** 3 good
**Rating:** 6
**Confidence:** 3

**Summary:**

The paper analyzes the covariance matrix of transformer networks at initialization, which commonly serves to analyze the training stability of deep networks. Specifically, the limiting distribution at network initialization is characterized by an SDE in the infinite depth-and-width limit regime. To enable the analysis, the authors make minor modifications to the attention and MLP formulation, which do not seem to impact their functionality much. Finally, promising preliminary numerical results are presented, which indicate that the transformer network with modified attention and MLPs can train stably and produce non-trivial performance without normalization layers.

**Strengths:**

The paper is well written and relatively easy to follow. The analysis looks solid as far as I can tell. In particular, both the self-attention and FFN blocks, both with skip connections, are analyzed, and only relatively minor modifications are made to make the analysis tractable. The paper also presents promising experimental evidence that the proposed shaped attention can be trained stably without normalization layers.

**Weaknesses:**

I appreciate the analysis on BERT-style language modeling in the appendix. It would be great to train the model with a comparable setup to BERT-Base and evaluate it on GLUE. That would then be a first piece of evidence that the proposed modified could be deployed in practice.

A curious thing is that the shaped attention in Eq. (9) can have negative values and values larger than 1, and no longer consists of proper PMFs. While this might not be an issue in practice, it is still a deviation from the principled formulation of the attention being normalized to 1. Can the authors think of an alternative formulation for which the normalization proprerty would still hold (or at least for which negative attention values could be avoided)?

Apart from that, I only see minor weaknesses. For example that the analysis is only for the network at initialization, but I agree with the authors that this can be addressed in follow-up work.


Typos: Missing space in L25 (“train. One”), L122 (“important properties”)


**Questions:**

* Could the proposed analysis be applied to decoder models with causally masked attention? Would masking the shaped attention lead to issues?
* Could the dependence on the average token discussed in L264 be an artifact of the bias towards diagonal values in Eq. (9)? Are there potential issues with long-range dependencies when introducing this bias?
* Assuming the authors used the embedding dim for the MLP hidden dim (both equal to 200), how would the results change when the depth, width and MLP hidden dim would be chosen with ratios like e.g. in BERT base?
* Besides trainability without normalization layers, did the authors observe other signs of improved stability? For example fewer spikes in the loss curve or lower training loss variance?


**Limitations:**

There are no major limitations, please see “Weaknesses”. The authors are upfront that they only present an analysis for the network at initialization, and that more sophisticated analyses should be done in follow-up work.

I’m not an expert in SDE-based analysis of neural networks and did not check the proofs in depth.

---

> ### Author Rebuttal · Authors · 2023-08-10
>
> Thank you for the positive and in-depth review!
>
> There are a number of ways to motivate this work. One is to come up with practically useful modifications. But our principal motivation was to study Transformers using the approach of proportional limits. However, the Transformer architecture does not scale in a stable way to infinite depth, so we needed to modify the architecture to one we can actually study using this technique. The modification should also ideally not hinder training performance.
>
> What we find (in Appendix D) is that, on some small benchmarks, we obtain performance comparable to standard architectures. We recognize that a much more comprehensive empirical study would be required to convince the community that the architecture is ready for deployment, but it is beyond our budget and it was not our main objective. We believe that the experiments we run is ample empirical evidence that our modification is at least worth studying. Based on your suggestion to look at GLUE and some further experiments that look at gradients at initialization, we think there's even some evidence that our modification might eventually be useful to practitioners, but that's very much a cherry on top, and we think even our preliminary results should generate excitement that we've found an architecture worth studying in the proportional limit.
>
> We now aim to address your remaining concerns. We hope that our response will be sufficient to view our work in an even more positive light.
>
> ## GLUE Benchmark
>
> We evaluated our pre-trained models on three small tasks from GLUE (due to time and resource constraints), and have included the results in Table 2 of our one-page pdf file. Overall, we found that the shaped Transformer performed slightly better on these three tasks than our baseline (best pre-LN Transformer). We also found these scores comparable to or outperforming the one-GPU-day-of-BERT-training results reported in [1]. However, we reiterate that the fairest comparison is against our own baselines, that match exactly the training procedure (data, number of pretraining steps, optimizer, etc..) that we adopt.
> Although these experiments are preliminary, we think this is an exciting signal that the shaped Transformers are worthy of a comprehensive empirical evaluation at larger scale.
>
> ## Shaped Attention Not Proper PMFs
>
> Indeed, PMFs are not required in practice for intermediate layers. However, we understand that having PMF might be a requirement in certain applications. We would kindly like to bring to your attention an experiment recovering the original Softmax (Eq. 33, Appendix D). Essentially, we initialize the Transformer in the shaped setting, but linearly decay the centering terms of the Softmax after 4000 steps of training, thus recovering a vanilla Transformer without layernorm. Hence, the output of the attention block becomes a proper PMF once again.
>
> The simulations depicted in Figure 3 show that we cannot simply remove centering, as otherwise depth quickly leads to degenerate correlations and covariance. Treating our shaped architecture as an initialization and letting the shaping parameters evolve through training is an exciting direction for future work.
>
> ## Analysis After Initialization
>
> So far, no work has been able to describe training dynamics in the proportional limit and this is one of the major open problems in the area. This is one of our longer-term goals and we are laying the foundations here: our results are analogous to the derivation of the NNGP in the infinite width limit. The NNGP was a key component of the NTK theory of training, which led to feature learning limits.
>
> ## Masking Shaped Attention
>
> Yes, we can extend shaped attention to the masked setting. The only modification will be for the centering matrix, which originally is the matrix of all $1/m$ entries, but would now have row $k$ taking the value of $1/k$ in the first $k$ entries, and 0 elsewhere.
>
> ## Dependence on the Average Token
>
> The average token term actually arises as a consequence of normalizing the Softmax output to $1$. If, for example, we were to replace the nonlinearity with say entry-wise sigmoids, then the average token terms would not show up in the Taylor expansion in Eq 8.
>
> At the same time, we do not see potential issues with long-range dependencies, as we still perform attention to all the tokens, and any attention matrix can be learned through optimization (thus, long-range dependencies can still be learned and strengthened through training).
>
> ## Varying Embedding Dimension
>
> Yes, we can handle general, non-uniform widths in the network, as each layer’s width $n_\ell^{-1}$ can be interpreted as a non-uniform step size. In this case, the total time (originally the depth-to-width ratio) would be replaced by the sum $\sum_\ell n_\ell^{-1}$. See earlier work by Hanin and Nica [2,3] which handled this general case similarly.
>
> ## Improved Training Stability
>
> First, we would like to point to our additional one-page PDF of Figures and Tables for plots on gradient norms. In Figure 1, gradients of the query/key parameters at initialization are vanishing in existing Transformer architectures, albeit milder with pre-LN. The *shaped Transformers do not suffer at all from vanishing gradients*.
>
> We also observed a couple of other signs of improved training stability:
>
> 1. When trained with Adam and learning rate warm-up, it appears that the shaped Transformer can handle larger learning rates without instability. In Figure 5 of the paper, we can see that unshaped Transformers are unstable at learning rate 0.001, while the shaped Transformer remains trainable.
>
> 2. The entropy does not seem to collapse as much for shaped Transformers, i.e. the Softmax does not concentrate on one token (Figure 6). The collapse of entropy is known to be correlated with unstable training [4], which the shaped Transformers seem to avoid during training.

---

> > ### Comment · Reviewer_h41x · 2023-08-14
> >
> > Thanks to the authors for providing a detailed rebuttal. I read it and also went over the other reviews and rebuttals.
> >
> > I appreciate the additional experiments on NLP eval sets. I believe follow-up work showing shaped attention in a transformer with standard transformer shape (such as depth, and width vs MLP hidden dimension scaling) and model capacity leading to strong empirical performance and improved scaling with depth could convince the community to more broadly adopt such an architecture.

---

> > > ### Author Response · Authors · 2023-08-16
> > > **Response**
> > >
> > > Thank you for your response.
> > >
> > > We would appreciate it if you could clarify one point for us. In your response, you say that you believe, if more extensive benchmarks show the same strong performance we've reported in our own experiments, we could convince the community to broadly adopt our new architecture. Given that you seem to continue to have some reservations about the work (since our rebuttal seems to not have improved your view of our paper), we were wondering if your hesitation to recommend acceptance, without equivocation, would have required us to have already run these extensive benchmarks, and be in a place to likely have our architecture broadly adopted by the community?
> > >
> > > If that's not the criterion for you to feel comfortable recommending this work for publication, without equivocation ("weak accept"), perhaps you can help us isolate the remaining sources of your concern. Besides not running large scale experiments, was there any other aspect of rebuttals that did not address your concern/question?

---

> > > > ### Comment · Reviewer_h41x · 2023-08-18
> > > >
> > > > After the rebuttal the description "Technically solid, moderate-to-high impact paper, with no major concerns with respect to evaluation, resources, reproducibility, ethical considerations." which is associated with "weak accept" still reflects my assessment well. I personally would need stronger empirical evidence at scale that modifications such as a non PMF attention do not have negative impact compared to the standard transformer architecture to justify "moderate-to-high impact on more than one areas".

---

### Official Review · Reviewer_5C73 · 2023-07-03

**Soundness:** 4 excellent
**Presentation:** 3 good
**Contribution:** 3 good
**Rating:** 6
**Confidence:** 3

**Summary:**

This paper studies a transformer architecture in the infinite width and depth limit where the ratio between the two remains constant. It is shown that, at initialization, the dynamics of the sample covariance matrix as it propagates through the layers converges to an SDE in the limit, where the SDE is indexed by the depth to width ratio. Moreover this limit is non-commutative and randomness preserving. The authors then propose the shaped attention - by constructing the attention matrix to resemble the identity function, and by properly scaling the residual connections and the softmax temperature, a more stable SDE is reached at the limit, theoretically allowing for more stable training.

**Strengths:**

1. The paper presents a rigorous analysis of the infinite depth-width limit, showing convergence of the dynamics (propagation of the data covariance at initialization) to a stochastic differential equation.
2. The proposed shaped attention architecture is intriguing, meriting further investigation as to its potential benefits.


**Weaknesses:**

My main issue with the paper is the lack of any convincing experimental result on the potential benefit of the proposed architecture. While it seems reasonable that a more stable forward process would lead to a more stable training and better performance, this is not always the case in practice, and indeed the paper barely makes any effort to confirm this. The experiment that is presented in section D of the appendix is in my opinion not convincing for the following reasons: 1) only 2 random seeds are considered for generating the results which is not close to enough, and 2)  the relative improvement is inside the standard deviation anyway (for d=24).  Without further experimental confirmation, i'm afraid the entire motivation and analysis is shaken to the point of being pointless from a practical perspective (I am willing to increase my score if such evidence is presented).
Having said that, i do not think that an immediate practical application is mandatory for any paper, however given the abundance of literature on the subject of infinite width/depth limits i see the contribution of the paper at its current form as incremental, more fitting a workshop than the a main conference contribution.

**Questions:**

Is there any reason to prefer an SDE limit over an ODE one (assuming a feature learning limit rather than the NTK)?

**Limitations:**

Limitations are adequately addressed

---

> ### Author Rebuttal · Authors · 2023-08-09
>
> We appreciate your constructive feedback and, in particular, your recognition of our rigorous theoretical analysis.
>
> Your chief criticism regards "the lack of any convincing experimental result on the potential benefit of the proposed architecture [to training in practice]". Besides your concern that we don't use enough seeds (we've since run more seeds), it seems you were hoping for our new architecture to outperform the standard Transformer architecture and that, without this demonstration, the entire motivation and analysis is "pointless from a practical motivation".
>
> Below we argue that you should adopt a different standard when judging our empirical work, and that our empirical findings are above and beyond what we might hope for when pursuing a proportional limit theory.
>
> ##  Aside: no abundance of work on proportional limit
>
> First, we would like to quickly address the idea that there is an "abundance of literature on the subject of infinite width/depth limits". This is true of the infinite *width* limit, and we're only aware of one novel study of the Transformer in this limit. On the other hand, there are, to our knowledge, only 3 papers on the proportional infinite depth-and-width limit that can handle more than one data point. Moreover, the proportional limit holds promise based on empirical evidence [3,5] that it leads to a much more accurate model of finite networks. Note that no work has yet succeeded in giving a concise description of training dynamics in the proportional limit and this is a major open problem. In this context, our work is the first to identify a Transformer-like architecture that admits a nondegenerate proportional limit.
>
> ## Role of empirical evaluation relates to motivation
>
> There are a number of ways to motivate this work:
>
> 1. A step towards a proportional limit model of Transformers and a theory for training dynamics. This is the principal motivation.
>
> 2. An architectural modification to address known instability at initialization of Transformers at greater depth. This is a necessary step to achieve #1.
>
> 3. Same as #2 but for improving training of Transformers at greater depth. This is a cherry on top.
>
> Limit theories have been immensely successful at deriving tractable models of neural networks. However, infinite-width theories have their limitations. This and related work on the proportional limit is pushing to derive more accurate models.
>
> The standard Transformer architecture, up to our knowledge, does not admit a stable proportional limit. It is unstable due to "degenerate signal propagation" phenomena. Here is the key: *we must modify the Transformer architecture in order to use proportional limits to study Transformer-like architectures.*
>
> Given that it is *necessary* to find a stable-at-depth Transformer, we go looking for one. Our paper can be interpreted as a rigorous study of the degeneracy, leading us to a modification that solves the signal propagation problem in the forward pass and admits a proportional limit.
>
> Is any such modification equally good? Clearly no, because we might have solved the stability problem but created a useless architecture. Is our architecture useless? Clearly no. In fact, Appendix D offers an exciting realization: *the Shaped Transformer seems to train just as well*, at least at the scales we were able to evaluate. (Several better-funded labs have contacted us to ask questions and are launching internal, independent efforts to evaluate our ideas at GPT scale. This is the way open science can accelerate advances, in our opinion.)
>
> ## Experiments
>
> In light of the above, we never needed to beat unshaped Transformers: We wanted to find an architecture that *could* be studied in the proportional limit and was *worth* studying. In fact, we might expect better results with tuning. At greater scale and depth, we expect degeneracy to affect the standard architecture more and more, suggesting the potential for even more significant improvements.
>
> In fact, thanks to your and the other reviewer’s suggestions, we are now in possession of more positive evidence. Results in our 1 page PDF suggest that shaping the attention model was *sufficient for a stable backward pass*, despite our theory addressing a stable forward pass. As you can see in Figure 1 of the one-pager, unshaped Transformers suffer significantly from vanishing gradients when depth is large in the query/key parameters. Meanwhile, *shaped Transformers do not have vanishing gradients at all*.
>
> Further, at the request of reviewer h41x, we report performance on a few tasks from the GLUE benchmark. We find that, compared to the baseline of unshaped Transformers, the shaped Transformers perform slightly better than the pre-ln baselines (see answer to reviewer h41x).
>
> In closing, the fact that we matched the performance of well-tuned Transformers, without LayerNorm and without re-tuning the optimization, is an exciting result. Our new results in the one-pager offer some even more exciting evidence that this architecture could be useful. Regardless, we think the architecture is clearly interesting enough to study in the proportional limit.
>
> ## SDE vs ODE Limits
>
> There is currently no significant evidence favoring either SDE or ODE limits. That being said, if we look at finite networks at scales that match the ODE 1/sqrt(depth) regime of [6], the SDE limit captures real randomness in the feature covariance matrix, which the ODE limit predicts is deterministic (see red curve in Figure 2). So we can conclude SDEs are better models here.
>
> We believe the judgment will be determined by future results, such as which limit serves as a more faithful model of finite size networks during and after training, which are impossible to speculate on at the moment. At this time, we believe both directions are well-motivated, and should be developed concurrently and in a complementary manner.
>
> We kindly hope you will be persuaded to reassess our work.

---

> > ### Author Response · Authors · 2023-08-16
> > **Response**
> >
> > We'd love to engage with you. Please let us know if you have any questions.

---

> > ### Comment · Reviewer_5C73 · 2023-08-18
> >
> > Thank you for your response.
> >
> > After reading the response of the authors and the other reviews, i am convinced this paper passes the bar for a theoretical contribution with possible practical applications, as indicated by the additional experiments. I will therefore increase my score accordingly.

---

### Official Review · Reviewer_4kLr · 2023-07-05

**Soundness:** 4 excellent
**Presentation:** 3 good
**Contribution:** 3 good
**Rating:** 7
**Confidence:** 4

**Summary:**

This paper investigates the convariance matrix of a modified softmax-based attention model with skip connections in the proportional limit of infinite-depth-and-width. They develop a theory to describe  the limiting distribution of the convariance matrix using a SDE indexed by the depth-to-width ratio. Based on this result, they propose a shaped attention to achieve a well-defined limit and ensure the stablity of the network.

**Strengths:**

1 The authors derive neural convarince SDEs for ReNets and softmax-based attention, respectively. Then, they combine the attention and residual layers, and give the SDE for full transformer architeture. To the best of my knowledge, the theoretical contributions are novel. The analysis technique is standard and the proof is convincing.

2 The theoretical contribution is also useful to guide the hyper-parameter tuning. The idea of shaped attention is very interesting, and the numerical simulations demonstrate the effectiveness of the proposed method.

**Weaknesses:**

1 All simulations are conducted at random initialization. It is better to conduct several experiments to show how the proposed initialization affect the training dynamics.

2 There are several typos, e.g.,  lines 25, ''.One'' -> ''. One'', line 122 , ''importantproperties"-> ''important properties", line 242 "remains -> remain".

**Questions:**

Please see weaknesses.

**Limitations:**

The authors have adequately addressed the limitations, and there is no potential negative societal impact of their work.

---

> ### Author Rebuttal · Authors · 2023-08-09
>
> Thank you for your positive and constructive review. As the reviewer correctly points out, our research has a dual impact on the realms of deep learning. From a practical perspective, it guides architecture design (and hyperparameter tuning). From a theoretical perspective, it is a step towards a more precise theory of realistic neural networks, as our SDE is the first precise characterization of a Transformers covariance at initialization.
>
> To begin, we will, of course, make our best effort to address all typos. Let us now address your main remark.
>
> ## Simulations
>
> With regards to simulations beyond initialization, we would like to first point to Appendix D, where we presented training results, beyond initialization, demonstrating that the modified (shaped Transformer) architecture (which can be scaled to infinite depth) trains to comparable accuracy as the standard (unshaped Transformer) architecture (which cannot be scaled). We stress that our architecture is freed of normalization layers.
>
> Your comment inspired us to offer more details on training dynamics, available in our "one-pager" (one-page PDF response) available to you. In the one-pager, we have simulated the gradient norm at initialization throughout different layers. For the queries and keys' parameters, we observe that while post-ln Transformers suffer from vanishing gradients, also the pre-ln Transformer's gradients severely diminish in magnitude with depth. This additional plot complements Figure 1 (left) of the paper in the sense that a high correlation between tokens in the forward pass is linked to vanishing gradients (see also [8]). On the other hand, the shaped Transformer has stable gradients, in the sense of depth-independent gradient magnitudes.
>
> Note that, in addition to our pre-training experiments in Appendix D mentioned above, we have added results for more seeds in our one pager. Just as in Appendix D, in these additional runs of our original training experiments, the shaped Transformer again matches the performance of well-tuned pre-ln Transformers. Note that we have not attempted to tune the training of the Shaped Transformer, so we might expect even better results given proper tuning, especially at greater depths, when we expect more instability for standard architectures. Regarding the training dynamics, we observe a higher trainability of our model with respect to the learning rate. This might be due to more resilience (in comparison to pre-ln Transformers) to entropy collapse, i.e. the regime where the softmax attends to only one token, in which case the network becomes untrainable (see Figures 5 and 6).
>
> In summary, we believe that, combining our existing experiments in Appendix D and the new evidence presented in our one-pager (including the additional evaluation on GLUE), we have addressed the weakness you have raised. We hope you might reassess the potential impact of our paper accordingly.

---

> > ### Comment · Reviewer_4kLr · 2023-08-11
> > **Replying to Rebuttal by Authors**
> >
> > The authors have addressed most of my concerns. Solid contribution.

---

### Official Review · Reviewer_J5pd · 2023-07-15

**Soundness:** 4 excellent
**Presentation:** 4 excellent
**Contribution:** 3 good
**Rating:** 7
**Confidence:** 4

**Summary:**

**Please note that this is an emergency review, completed in a limited amount of time whilst I am traveling.**

This paper studies the properties of a class of attention-based networks, modified from the standard Transformer, in the limit in which depth and width tend proportionally to infinity. Technically, it extends recent work by Li, Nica, and Roy (NeurIPS 2022) on SDE-based descriptions of MLPs with random weights. The central result is a modification of the attention mechanism, referred to by the authors as shaped attention, that gives a non-trivial limit.

**Strengths:**

In my abbreviated reading, I found the paper clear and enjoyable to read. I think it is a worthy contribution to the rapidly-growing literature on double-scaling limits of deep and wide networks, as it takes a substantial step closer to the architectures used in practice relative to prior works. The non-commutative scaling regime of ResNets studied by the authors nicely compliments recent work by Hayou and Yang on commutative infinite-width and -depth limits of residual networks. I found the observations regarding finite-time numerical instabilities to be particularly interesting.

Due to time constraints, I have not had the chance to go through the proofs in detail.

**Weaknesses:**

To my mind, the main weakness of this work is that it does not address inference. However, this reflects the larger issue that it is challenging to perform inference in conditionally-Gaussian processes (perhaps outside the nearly-Gaussian limit, where things can be studied perturbatively). Thus, this limitation does not significantly dampen my enthusiasm for the authors' work.

**Questions:**

- Lines 46-47: In discussing the perturbative regime, work by Dyer and Gur Ari (2019) and by Zavatone-Veth, Canatar, Ruben, and Pehlevan (2021) should be cited alongside the work of Roberts, Yaida, and Hanin (2021).

- Lines 48-53: For completeness, it might be useful to cite work on the proportional limit from random matrix theory, e.g. by G. Akemann and colleagues.

- Line 243: "Covariane" -> "Covariance"

- Lines 322-333: In relation to my comment about inference above, I think this paragraph would be substantially improved if the authors could offer a more concrete suggestion of how their approach could be leveraged to study training and generalization.

- Lines 381-382: Why is the arXiv version of Li, Nica, and Roy cited in place of the NeurIPS 2022 publication?

**Limitations:**

I think the authors adequately discuss the limitations of their work.

---

> ### Author Rebuttal · Authors · 2023-08-09
>
> ## Summary
>
> Thank you for your positive and constructive review.  We agree with the reviewer on the importance of studying the proportional limit for realistic architectures such as ResNets and attention models, as it paves the way for a faithful theory of neural networks beyond the infinite width limit. This is indeed one of the main motivations of our work. We also thank you for acknowledging the relevance of our results on resnets (in relation to Hayou and Yang), and for enjoying the analysis of the SDE’s stability. We will add or fix the citations you mentioned in the review.
>
> In your review, you suggest a weakness of this work is that we do not develop inference. We would like to gently push back, and suggest that inference is ancillary to our goals, and that our work represents a key first step to building a more faithful model of training.
>
> ## Inference missing?
>
> The derivation of the NNGP in the infinite-width setting generated interest in the possibility of inference in this structure. In much the same way, as you've noticed, our work motivates inference in the conditional GP defined by the proportional limit. While we think inference in the NNGP was interesting, it is not our goal to study inference, so we do not view it as a limitation.
>
> Indeed, we believe that, looking back, the role of our work will be in defining boundary conditions for the yet-to-be-characterized processes describing training (by gradient methods), offering a model for training realistic networks that might offer us some theoretical insight, if it is tractable enough. This work is thus a first step towards understanding training.
>
> That said, we did consider what inference might entail. Specifically, how might we sample an output at a specified input, conditional on a set of data points $\{ x^\alpha, y^\alpha \}_{\alpha=1}^m$. We agree with you that this is not a trivial task. It is also orthogonal to our contributions.
>
> When we imagine the conditional diffusion dynamics, we expect this will create a new drift term in the SDE via Doob’s h-transform, which is not always easy to compute, since it can depend on the time marginals. In fact, this drift is closely related to score functions in diffusion models, which are known to be difficult to compute. After publication, we will be happy to discuss ideas around inference with any interested parties.
>
> ## Training and Generalization
>
> For future directions, we can consider the roadmap taken by infinite-width theories up to this point. For example, the calculations required for the Neural Tangent Kernel (NTK) are essentially identical to those of the Gaussian process results. While there are some technical difficulties, it is not unreasonable to foresee a similar SDE result for the NTK. This will immediately lead to calculations for the first step of training, which already allows to study which regimes allow for feature learning and learning rate transfer.
>
> Generalization however, will require knowledge about the complete dynamics of training. This is currently not known for three-layer neural networks in the feature learning regime, and therefore we do not foresee infinite-depth limits to be any easier. The closest approach is the self-consistent dynamical mean field theory (DMFT) equations of Bordelon and Pehlevan [7], which we believe is possible to extend to infinite-depth with more careful calculations. Either way, it remains unclear how we can characterize the end of training, which is required for generalization results, but we suspect the high-level roadmap will serve as a helpful guide nonetheless.

---

> > ### Comment · Reviewer_J5pd · 2023-08-10
> >
> > I thank the authors for their thoughtful and thorough reply to my questions and those of the other referees. I remain solidly in favor of acceptance.

---

### Author Rebuttal · Authors · 2023-08-09

We want to thank all the reviewers for your detailed and constructive reviews. In particular, we appreciate the overwhelmingly positive feedback towards our theoretical results. Our work is the first key step towards a faithful theory of Transformer-like architectures. Of course, to study these architectures using proportional limits, we needed to develop architectural modifications that produced stable signal propagation. From the perspective of theory building, a merely trainable architecture would be an exciting outcome, as we would have arrived at a Transformer-like architecture worth studying via proportional limits.

That said, we think it's quite exciting that our preliminary experiments find the Shaped Transformers train equally well as fine-tuned, unshaped Transformers on small (in modern terms) benchmarks. These results suggest that with further fine-tuning, especially in deeper regimes where the signals are even more degenerate, we believe the Shaped Transformer has the potential to improve even further. Still, we think these findings are cherries on top of what is a solid theoretical contribution worth sharing with the community.

To provide some further evidence that the architecture might not be merely interesting but also useful, we have summarized further simulations and experiments in our one-page pdf:

Figure 1. We measure the gradient norm of both shaped and unshaped Transformers at initialization, and find that shaping the attention mechanism successfully prevents gradients from vanishing, while unshaped Transformers suffer from rapidly vanishing gradients.

Table 1. As requested by a reviewer, we have added more seeds (2 more, hence 4 in total) to our training experiments, replicating the same results as we already reported in Appendix D.

Table 2. We have added experiments on three datasets from the GLUE benchmark. These demonstrate that our shaped Transformers outperform our pre-ln baseline and even previously reported one-GPU-day results. We plan to expand GLUE results, though we would like to reiterate that we do not need to outperform existing architectures to have identified an interesting model to study.

To summarize, we have demonstrated that the shaped Transformer has stable gradients at initialization and throughout training, and can match the performance of well-tuned unshaped Transformers with the same settings on some small benchmarks.

# Experiment Details

Figure 1. We take the same hyperparameters as Figure 1 of the submission, but with width $n=256$ and use causally masked attention for an autoregressive task predicting code data. In this setting we have sequence lengths of 128, and the error bars in the plot correspond to one standard deviation of each parameter matrix’s gradient norm, over a batch of 32 sequences.

Table 1. We add 2 more seeds for each experiment and again report the best-performing model across the considered learning rates, as explained in the paper. We have also increased the number of datapoints for the evaluation of the training statistics and re-performed the evaluation.

Table 2. We take the checkpoints of the baselines and selected shaped transformers ($\gamma=0.2$ for the shallower model $d=18$, $\gamma=0.05$ for the deeper model $d=24$). Then, as per standard, we replace the final head with a randomly initialized MLP with 1 hidden layer and tanh activation function. We then fine-tune on each of three datasets from GLUE (COLA, MRPC, RTE) for 5 epochs and Adam optimizer $lr=5e-4$ with batch size 16, and report test evaluation metric for the corresponding task.

# References

1. Geiping and Goldstein. "Cramming: Training a Language Model on a single GPU in one day." (2023).

2. Hanin and Nica. "Products of many large random matrices and gradients in deep neural networks." (2020).

3. Hanin and Nica. "Finite Depth and Width Corrections to the Neural Tangent Kernel." (2019).

4. Zhai, Likhomanenko, Littwin, Busbridge, Ramapuram, Zhang, Gu, and Susskind. "Stabilizing Transformer Training by Preventing Attention Entropy Collapse." (2023).

5. Li, Nica, and Roy. "The Neural Covariance SDE: Shaped Infinite Depth-and-Width Networks at Initialization." (2022).

6. Hayou and Yang. "Width and Depth Limits Commute in Residual Networks." (2023).

7. Bordelon, Blake, and Cengiz Pehlevan. "Self-consistent dynamical field theory of kernel evolution in wide neural networks." (2022):

8. Noci et al. "Signal propagation in transformers: Theoretical perspectives and the role of rank collapse." (2022)

---

### Decision · Program_Chairs · 2023-09-21

**Decision:**

Accept (poster)

**Comment:**

The paper provides an abstract model for the dynamics of Gram matrices after applying a modified version of self-attention in mean field regimes. Such abstraction allows leveraging tools from stochastic calculus to study the inherent bias of random self-attention. The paper is well written and theoretical analyses are well motivated.